# Reduced oxidative capacity in macrophages results in systemic insulin resistance

Saet-Byel Jung[1], Min Jeong Choi[1], Dongryeol Ryu[2,3], Hyon-Seung Yi[4], Seong Eun Lee[1], Joon Young Chang[1], Hyo Kyun Chung[1], Yong Kyung Kim[1], Seul Gi Kang[1], Ju Hee Lee[4], Koon Soon Kim[1,4], Hyun Jin Kim[4], Cuk-Seong Kim[5], Chul-Ho Lee[6], Robert W. Williams[7], Hail Kim[8], Heung Kyu Lee [8], Johan Auwerx[2] & Minho Shong[1,4]

Oxidative functions of adipose tissue macrophages control the polarization of M1-like and M2-like phenotypes, but whether reduced macrophage oxidative function causes systemic insulin resistance in vivo is not clear. Here, we show that mice with reduced mitochondrial oxidative phosphorylation (OxPhos) due to myeloid-specific deletion of CR6-interacting factor 1 (*Crif1*), an essential mitoribosomal factor involved in biogenesis of OxPhos subunits, have M1-like polarization of macrophages and systemic insulin resistance with adipose inflammation. Macrophage GDF15 expression is reduced in mice with impaired oxidative function, but induced upon stimulation with rosiglitazone and IL-4. GDF15 upregulates the oxidative function of macrophages, leading to M2-like polarization, and reverses insulin resistance in *ob/ob* mice and HFD-fed mice with myeloid-specific deletion of *Crif1*. Thus, reduced macrophage oxidative function controls systemic insulin resistance and adipose inflammation, which can be reversed with GDF15 and leads to improved oxidative function of macrophages.

[1] Research Center for Endocrine and Metabolic Diseases, Department of Medical Science, School of Medicine, Chungnam National University, Daejeon 35015, Korea. [2] Laboratory for Integrative and Systems Physiology, Institute of Bioengineering, École Polytechnique Fédérale de Lausanne, 1015 Lausanne, Switzerland. [3] Laboratory of Molecular and Integrative Biology, Department of Korean Medical Science, School of Korean Medicine, Pusan National University, Yangsan 50612, Korea. [4] Department of Internal Medicine, Chungnam National University Hospital, Daejeon 35015, Korea. [5] Department of Physiology, Department of Medical Science, School of Medicine, Chungnam National University, Daejeon 35015, Korea. [6] Laboratory Animal Resource Center, Korea Research Institute of Bioscience and Biotechnology, Daejeon 34141, Korea. [7] Department of Genetics, Genomics and Informatics, University of Tennessee Health Science Center, Memphis, TN 38163, USA. [8] Graduate School of Medical Science and Engineering, Korea Advanced Institute of Science and Technology, Daejeon 34051, Korea. Correspondence and requests for materials should be addressed to M.S. (email: minhos@cnu.ac.kr)

Adipose tissue macrophages (ATM) are abundant immune cells in human and murine adipose tissue[1,2]. The discovery of ATMs, and their elevated numbers in the adipose tissues of obese mice and humans, indicates mechanisms by which obesity induces adipose inflammation and systemic insulin resistance[1,3,4]. ATMs, which have substantial functional heterogeneity, actively participate in various aspects of nutrient metabolism within adipose tissue[4]. The function of macrophages in shifting adipose tissue inflammation toward a pathological state has attracted increasing attention. Alternatively activated (M2-like) macrophages, which typically express the mannose receptor CD206 and secrete anti-inflammatory cytokines, are located mainly in lean adipose tissue[5,6]. However, adipose tissues from obese individuals contain elevated numbers of classically activated (M1-like) macrophages, which produce proinflammatory mediators such as tumor necrosis factor α (TNF), monocyte chemoattractant protein-1 (MCP-1), and interleukin-1α (IL-1α)[7–9]. In this context, the signal transducer and activator of transcription 6 (STAT6) transcription factor, which is activated upon stimulation by type 2 cytokines such as IL-4 and IL-13, promotes maturation into alternatively activated macrophages[10]. Moreover, STAT6 and its associated transcription factors, including the peroxisome proliferator-activated receptor-γ (PPARγ), PPARδ, and peroxisome proliferator-activated receptor-γ coactivator-1α (PGC-1α), have critical functions in M2-like macrophages by increasing oxidative metabolism and mitochondrial biogenesis[11–13]. Oxidative metabolism in M2 macrophages has been directly linked to an anti-inflammatory phenotype[14]. Inhibition of fatty acid oxidation (FAO) in M2 macrophages with the carnitine palmitoyltransferase 1 (CPT1) inhibitor etomoxir limits M2 activation in response to IL-4[15]. Blocking oxidative metabolism not only impairs the development of an M2-like phenotype, but also drives the cells toward an M1-like state[16]. Furthermore, failure of alternative M2 activation, which is associated with reduced oxidative function, leads to classical macrophage activation, elevated weight gain, and obesity with concurrent adipose inflammation and insulin resistance[17,18]. However, it is unclear whether the oxidative function of macrophages is reduced under these conditions, and if so, whether that impairment induces adipose tissue inflammation and insulin resistance. Furthermore, it is not known whether a treatment capable of increasing the oxidative function of macrophages would reverse insulin resistance and adipose inflammation.

In this study, we show that mice with impaired mitochondrial oxidative function in macrophages due to myeloid-specific loss of function of the CR6-interacting factor 1 (Crif1) gene, an essential mitoribosomal factor required for biogenesis of oxidative phosphorylation (OxPhos) subunits, develop systemic insulin resistance associated with adipose inflammation. Moreover, macrophages from these mice are deficient in release of the secretory factor growth differentiation factor 15 (GDF15), which is required for oxidative metabolism in M2-like macrophages stimulated with IL-4 and the PPARγ agonist rosiglitazone. GDF15 expression is suppressed in M1-like macrophages, which promote adipose inflammation and insulin resistance. GDF15 also upregulates the oxidative function of macrophages, leading to polarization into an M2-like phenotype, and reverses insulin resistance in Crif1-deficient mice fed a high-fat diet (HFD). In addition, GDF15-deficient macrophages polarized into an M1-like phenotype, and reintroduction of GDF15-null macrophages into HFD-fed mice in which macrophages are depleted with clodronate, results in glucose intolerance. Furthermore, administration of GDF15 to obese mice improves the oxidative function of adipose macrophages and reverses insulin resistance. Collectively, our findings demonstrate that defective mitochondrial oxidative function in macrophages causes systemic insulin resistance and adipose tissue inflammation. In addition, macrophages with defects in mitochondrial OxPhos function lose the ability to secrete GDF15, a macrophage-regulating autocrine and paracrine signaling factor that promotes anti-inflammatory responses in white adipose tissue.

## Results

**Impaired oxidative function increases M1-like macrophages.** Although mitochondrial oxidative function has been shown to modulate macrophage activation, its in vivo effect on systemic insulin resistance has not been validated. Insulin resistance is associated with reduced M2-like polarization of macrophages[19]. To elucidate the crosstalk between immune programs of macrophage activation and mitochondrial OxPhos function, we generated myeloid cell-specific OxPhos complex-deficient mice by deleting Crif1 from myeloid cells, including macrophages, using the LysM-Cre driver[20]. Merging of endogenous CRIF1 fluorescence with that of MitoTracker revealed that CRIF1 was present in the mitochondria of macrophages (Supplementary Fig. 1A). We then generated heterozygous and homozygous myeloid cell-specific Crif1-deficient mice (MacHE and MacHO, respectively) and performed genotyping to confirm mutagenesis (Supplementary Fig. 1B). Bone marrow–derived macrophages (BMDMs) of MacHE and MacHO mice exhibited a significant and marked reduction in the Crif1 mRNA level in comparison with control mice (MacWT) (Fig. 1a). As expected, BMDMs of MacHO mice expressed lower levels of the OxPhos subunits NDUFA9, UQCRC2, COX4I1, and ATP5A1 than control mice (Fig. 1b, c, Supplementary Fig. 12).

We next examined the integrity of OxPhos complexes in macrophages using BN-PAGE. BMDMs of MacHE and MacHO mice expressed reduced levels of complexes I, II, IV, and V, and formed an abnormal supercomplex between complexes III and IV (Supplementary Fig. 1C). Monitoring the oxygen consumption rate (OCR) in BMDMs revealed that macrophages from MacHO mice exhibited reduced basal respiration, ATP-linked respiration, and maximal respiration rates (Fig. 1d, e). Deletion of the Crif1 gene from myeloid cells using the LysM-Cre driver did not affect the populations of monocytes or other myeloid cells within the blood and bone marrow (Supplementary Fig. 1D–1G). These findings indicate that Crif1 deficiency decreases the number and assembly of OxPhos subunits, as well as respiration, and may cause abnormal mitochondrial proteostasis.

To determine whether Crif1 deficiency plays a critical role in the switch between classically and alternatively activated macrophages, we examined the expression of M1-associated and M2-associated genes in the presence or absence of Th1 and Th2 cytokines. In the absence of any stimuli, BMDMs isolated from MacHO mice expressed higher levels of M1-related genes, and lower expression of M2-related genes, than controls (Fig. 1f). In addition, under basal conditions these cells expressed higher levels of Il6 and Nos2 mRNA levels than macrophages from MacWT mice, and these genes were induced even more strongly in response to interferon gamma (IFN-γ) and lipopolysaccharides (LPS) (Fig. 1g, h). Furthermore, IL-4–stimulated induction of Arg1 (L-arginase) and Ym1 (Chi3l3) was lower in BMDMs from MacHO mice than in those from MacWT mice (Fig. 1i, j). In addition, p38 kinase, which regulates inflammatory responses in M1-like macrophages[21], was activated to a greater extent by IFN-γ and LPS in MacHO than in MacWT (Fig. 1k, l, Supplementary Fig. 13). Taken together, these data suggest that macrophages with genetically impaired oxidative function have properties consistent with M1-like polarized macrophages and reduced Th2 cytokine responsiveness.

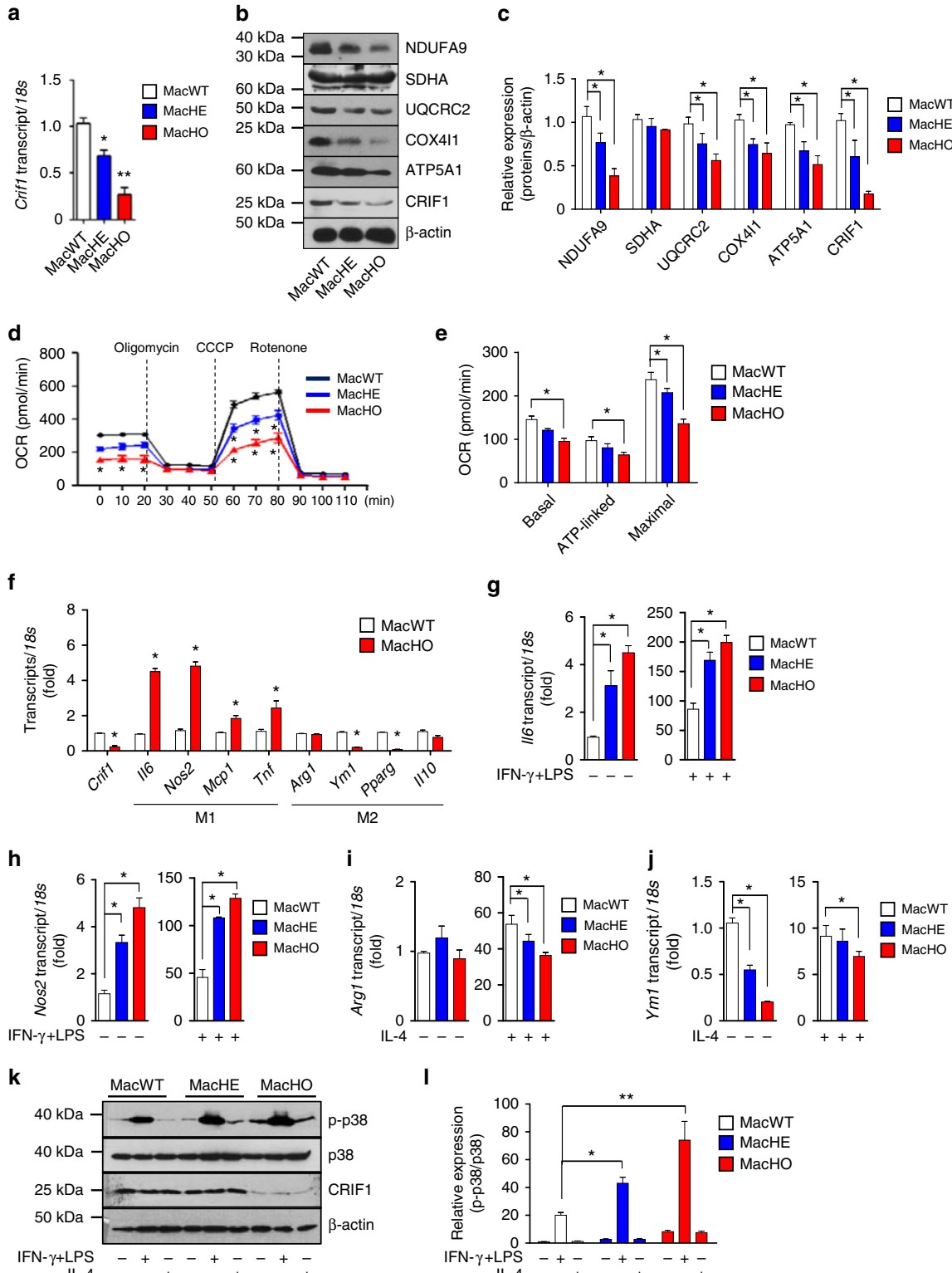

**Fig. 1** Oxidative dysfunction by CRIF1 deficiency increases M1-like macrophages **a** Real-time PCR analysis of *Crif1* in BMDMs from 8-week-old MacWT, MacHE, and MacHO mice. **b**, **c** Immunoblot analysis of OxPhos complex and CRIF1 expression in BMDMs from 8-week-old MacWT, MacHE, and MacHO mice. **d**, **e** Oxygen consumption rates (OCR) in BMDMs from 8-week-old MacWT, MacHE, and MacHO mice. CCCP: carbonyl cyanide m-chlorophenyl hydrazone. **f** Real-time PCR analysis of M1-related and M2-related genes in BMDMs from 8-week-old MacWT and MacHO mice. **g**, **h** Real-time PCR analysis of *Il6* and *Nos2* mRNA expression in the presence or absence of IFN-γ (10 ng/ml) and LPS (100 ng/ml). **i**, **j** Real-time PCR analysis of *Arg1* and *Ym1* mRNA expression in the presence or absence of IL-4 (20 ng/ml). **k**, **l** Immunoblot analysis of p38 phosphorylation after exposure to IFN-γ (10 ng/ml) and LPS (100 ng/ml) for 30 min. Data are expressed as means ± SEM and are representative of three independent experiments ($n = 5$ mice per group). $*p < 0.05$ or $**p < 0.01$ (two-tailed Student's *t* test)

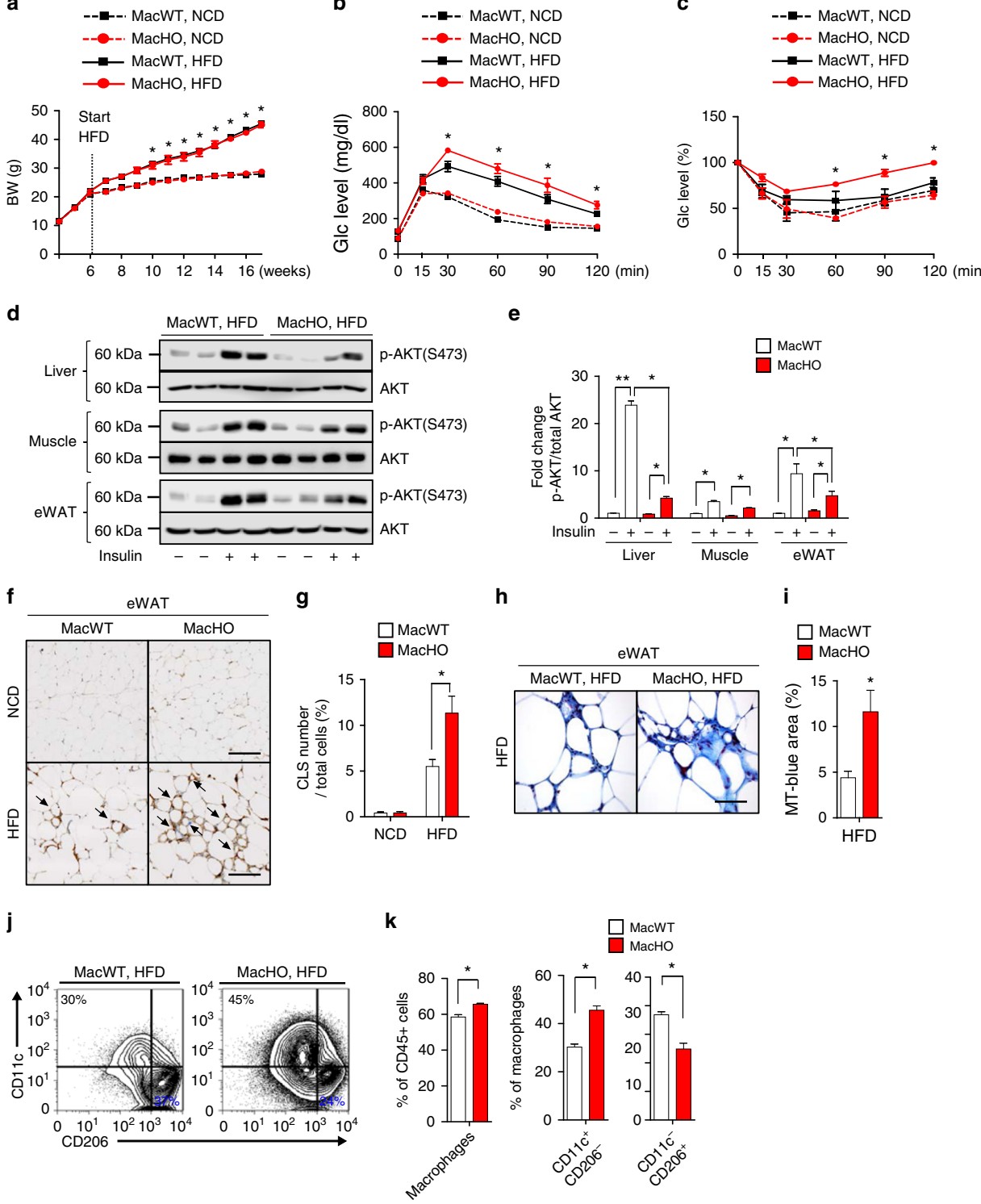

**Fig. 2** MacHO mice fed a HFD exhibit systemic insulin resistance. (**a**) Body weight of MacWT and MacHO mice fed a NCD or HFD for 10 weeks (MacWT, $n = 5$; MacHO, $n = 5$). **b, c** MacWT or MacHO mice fed a NCD or HFD were subjected to IPGTT after a 16 h fast and an insulin tolerance test after a 4 h fast (MacWT, $n = 5$; MacHO, $n = 5$). **d, e** Immunoblot analysis of AKT (S473) phosphorylation in liver, muscle, and eWAT 30 min after intraperitoneal injection of insulin into MacWT or MacHO mice fed a HFD. **f, g** Immunohistochemical analysis (staining with an anti-F4/80 antibody and hematoxylin) of macrophages in eWAT of MacWT or MacHO mice fed a HFD. Scale bar: 200 μm. **h, i** Masson's trichrome staining (for fibrosis) of eWAT from MacWT or MacHO mice fed a HFD. Scale bar: 50 μm. (**j** and **k**) FACS analysis of total macrophages (CD45$^+$/F4/80$^+$/CD11b$^+$) and M1 (CD11c$^+$/CD206$^-$) and M2 (CD11c$^-$/CD206$^+$) macrophages infiltrated into eWAT of MacWT or MacHO mice fed a HFD. Data represent means ± SEM and are representative of three independent experiments. *$p < 0.05$ or **$p < 0.01$ (two-tailed Student's $t$ test)

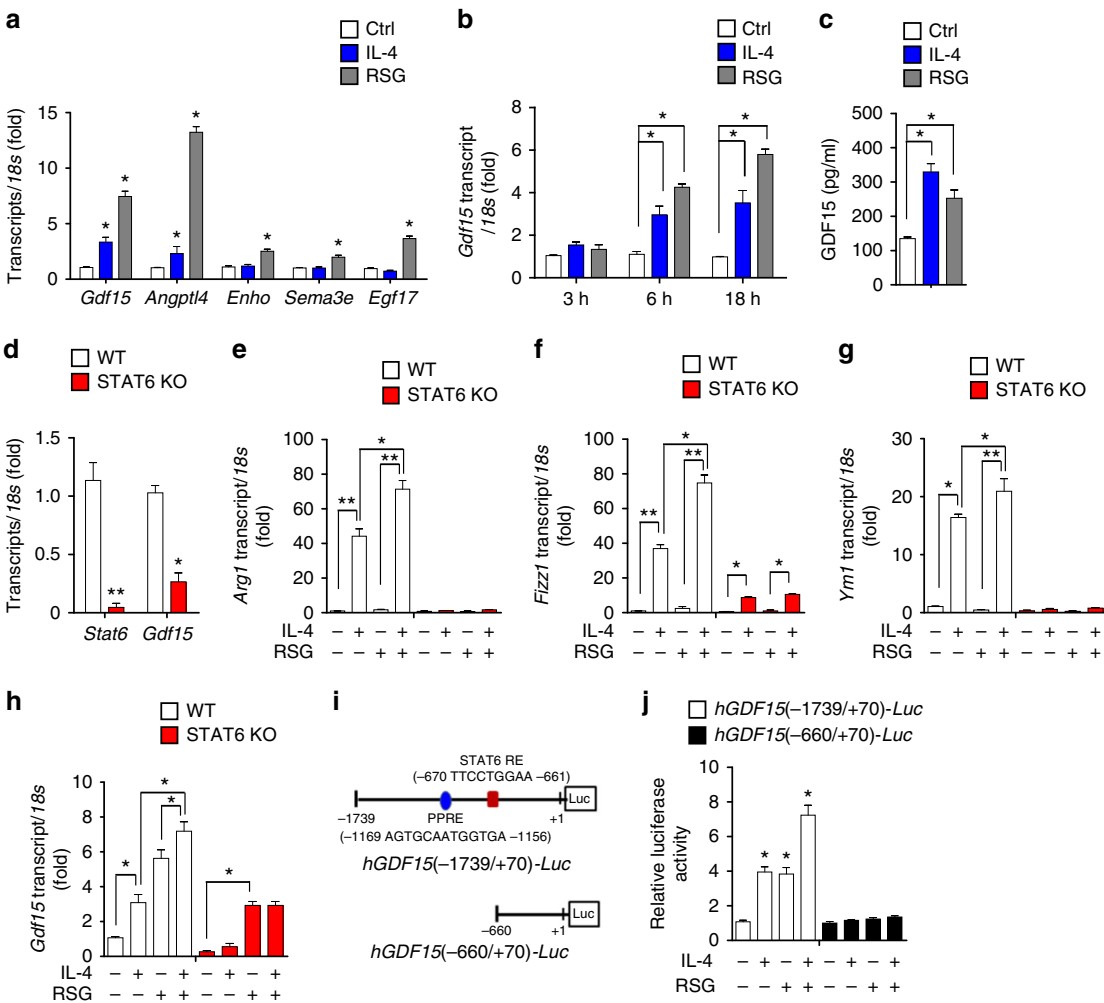

**Fig. 3** PPARγ agonist and STAT6 modulate GDF15 expression. **a** Real-time PCR analysis of *Gdf15*, *Angptl4*, *Enho*, *Sema3e*, and *Egf17* mRNA expression in BMDMs 18 h after treatment with rosiglitazone (RSG) (10 μM) and IL-4 (100 ng/ml). **b** Real-time PCR analysis of *Gdf15* mRNA in BMDMs 3, 6, and 18 h after treatment with rosiglitazone (10 μM) and IL-4 (100 ng/ml). **c** GDF15 levels in culture supernatants from BMDMs treated with rosiglitazone (10 μM) and IL-4 (100 ng/ml) for 48 h. **d** Real-time PCR analysis of *Stat6* and *Gdf15* mRNA expression in BMDMs from WT and *Stat6*-KO mice. **e–h** Real-time PCR analysis of *Arg1*, *Fizz1*, *Ym1*, and *Gdf15* mRNA expression in BMDMs after 18 h treatment with rosiglitazone (10 μM) and IL-4 (100 ng/ml) from WT and *Stat6*-KO mice. **i** Schematic structure of reporter constructs containing a 1739 or 660 bp fragment of the human *GDF15* promoter. **j** Relative luciferase activity induced by the human *GDF15* promoter in RAW264.7 cells transfected with *hGDF15(-1739/+70)-Luc* or *hGDF15(-660/+70)-Luc* and cultured with or without rosiglitazone (10 μM) and IL-4 (100 ng/ml) for 24 h. Data are expressed as means ± SEM and are representative of three independent experiments. *$p < 0.05$ or **$p < 0.01$ (two-tailed Student's $t$-test)

**MacHO mice fed a HFD develop systemic insulin resistance**. To determine whether the reduced oxidative function in macrophages triggers systemic insulin resistance, we monitored body weight and glucose tolerance in MacHO mice fed either a normal chow diet (NCD) or a HFD. We observed no difference in body weight or the levels of cholesterol or triglyceride between MacHO and MacWT mice fed a NCD or HFD (Fig. 2a, Supplementary Fig. 2A). In addition, we detected no significant differences between MacHO and MacWT in glucose and insulin tolerance under NCD. However, MacHO mice fed a HFD developed systemic glucose intolerance and insulin resistance (Fig. 2b, c). To confirm systemic insulin resistance in MacHO mice on a HFD, we examined AKT phosphorylation in liver, skeletal muscle, and adipose tissue. Phosphorylation of AKT in liver and epididymal white adipose tissue (eWAT) was clearly lower in MacHO than in MacWT mice, although no difference was detected in skeletal muscle (Fig. 2d, e, Supplementary Fig. 14). Although histological analysis of adipose tissue, liver, and skeletal muscle did not reveal any clear abnormalities (Supplementary Fig. 2B, 2C), HFD-fed

MacHO mice exhibited characteristic macrophage infiltration of eWAT, accompanied by an elevated number of crown-like structures (Fig. 2f, g). Furthermore, fibrosis, which is a sequela of significant long-term macrophage-mediated inflammation, was widespread throughout the eWAT of HFD-fed MacHO mice (Fig. 2h, i).

The proportion of macrophages within the CD45+ cell population among stromal vascular cells (SVCs) of eWAT was significantly higher in MacHO than in MacWT mice fed a HFD (65.65 ± 0.92% vs. 58.4 ± 2.28%, respectively; $p < 0.05$, Student's $t$-test) (Fig. 2j, Supplementary Fig. 3). The percentage of M1-like macrophages within the total macrophage population (F4/80+/CD11b+/CD11c+/CD206-) was significantly higher in MacHO mice than in MacWT mice (45.56 ± 3.06% vs. 30.26 ± 2.15%, respectively; $p < 0.05$, Student's $t$ test), whereas the percentage of M2-like macrophages (CD11b+/F4/80+/CD11c-/CD206+) was significantly lower (24.86 ± 3.53% vs. 36.86 ± 1.77%; $p < 0.05$, Student's $t$-test) in MacHO mice (Fig. 2j, k). Although the macrophage population changed, the proportion of eosinophils

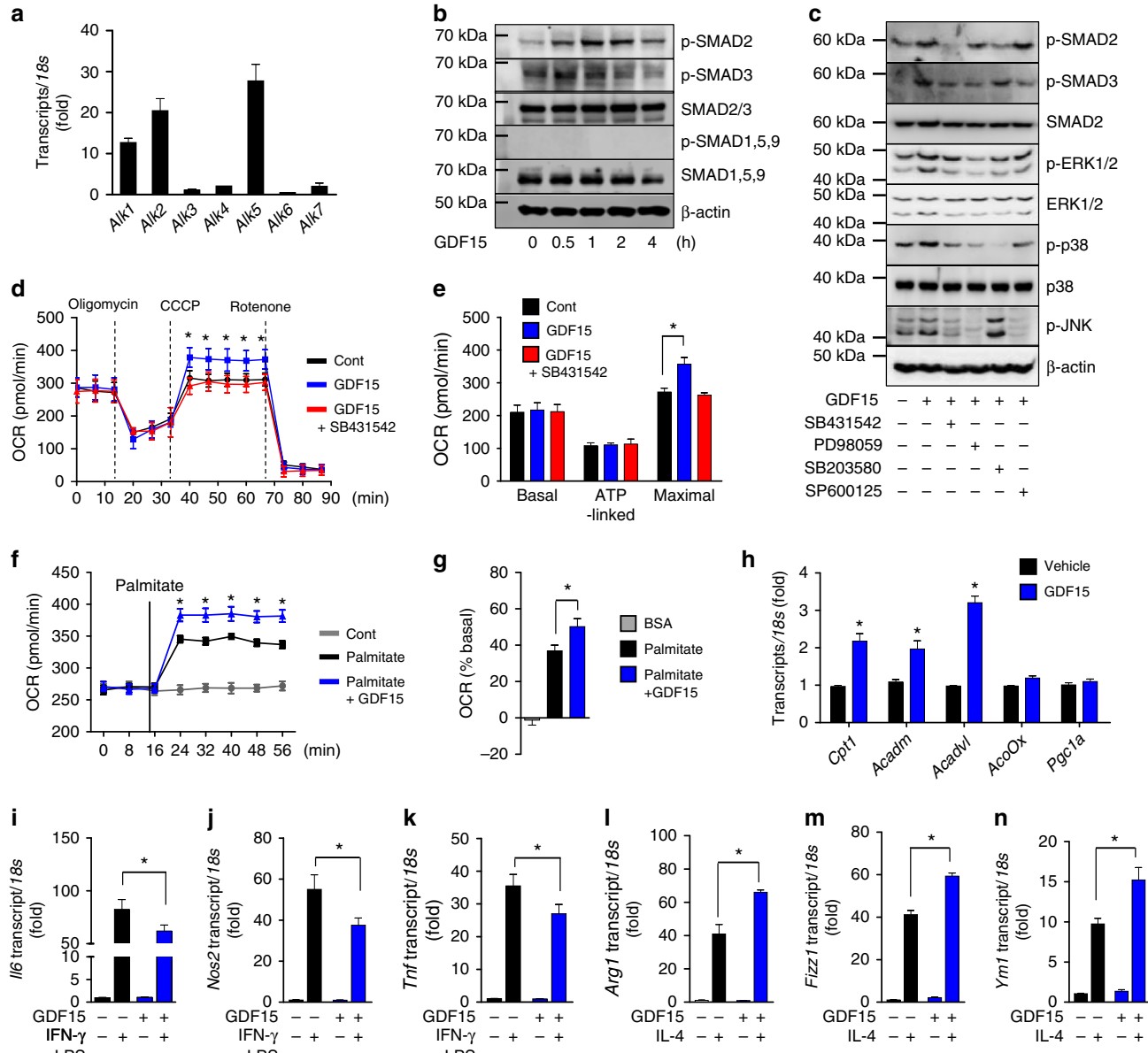

**Fig. 4** GDF15 promotes oxidative metabolism and M2-like polarization. **a** Real-time PCR analysis of TGF-β RI (*Alk1–7*) expression in BMDMs. **b** Immunoblot analysis of rGDF15-induced phosphorylation of SMAD2 and 3. **c** Immunoblot analysis of rGDF15-induced phosphorylation of SMAD2, SMAD3, ERK1/2, p38, and JNK, which was inhibited by the chemical inhibitors, SB431542 (an ALK4, 5, 7 inhibitor; 5 μM), PD98059 (an ERK1/2 inhibitor; 25 μM), SB203580 (a p38 inhibitor; 10 μM), and SP600125 (a JNK inhibitor; 10 μM). **d, e** OCR in BMDMs treated with rGDF15 (300 ng/ml) for 24 h with or without SB431542 (5 μM). **f, g** Fatty acid oxidation in BMDMs treated with rGDF15 (300 ng/ml) for 24 h with or without palmitate-BSA (200 μM). **h** Real-time PCR analysis of FAO-related genes in rGDF15 (300 ng/ml)-treated BMDMs. **i–k** Real-time PCR analysis of M1-related gene expression in rGDF15 (300 ng/ml)-treated BMDMs in the presence of IFN-γ (10 ng/ml) and LPS (100 ng/ml). (**l–n**) Real-time PCR analysis of M2-related gene expression in rGDF15-treated BMDMs in the presence of IL-4 (20 ng/ml). Data are expressed as means ± SEM and are representative of three independent experiments. *$p < 0.05$ or **$p < 0.01$ (two-tailed Student's $t$ test)

and CD4$^+$ and CD8$^+$ T cells did not differ between the two groups (Supplementary Fig. 2D). Taken together, these findings indicate that reduced oxidative function in macrophages triggers systemic insulin resistance, characterized by peripheral adipose tissue inflammation and a shift in macrophage polarity from M2 to M1 in adipose tissue.

**PPARγ and STAT6 regulate expression of GDF15 in macrophages.** Several independent lines of evidence support the importance of PPARγ-stimulated oxidative metabolism in priming and polarization of the alternative macrophage phenotype[11,22,23]. To identify the paracrine factors that contribute to macrophage polarization in adipose tissue, we first analyzed six transcriptomes from control human macrophages and macrophages treated with rosiglitazone, a potent PPARγ agonist (GSE25088; Supplementary Fig. 3A, 3B)[24]. Although we identified 12 humoral factors whose expression levels were induced by rosiglitazone (Supplementary Fig. 4A), only epidermal growth factor 17 (*EGF17*), semaphorin 3e (*SEMA3e*), angiopoietin-related protein 4 (*ANGPTL4*), energy homeostasis (*ENHO*), and Growth differentiation factor 15 (*GDF15*) were significantly upregulated (Supplementary Fig. 4B). Next, we examined the induction of these autocrine and paracrine factors

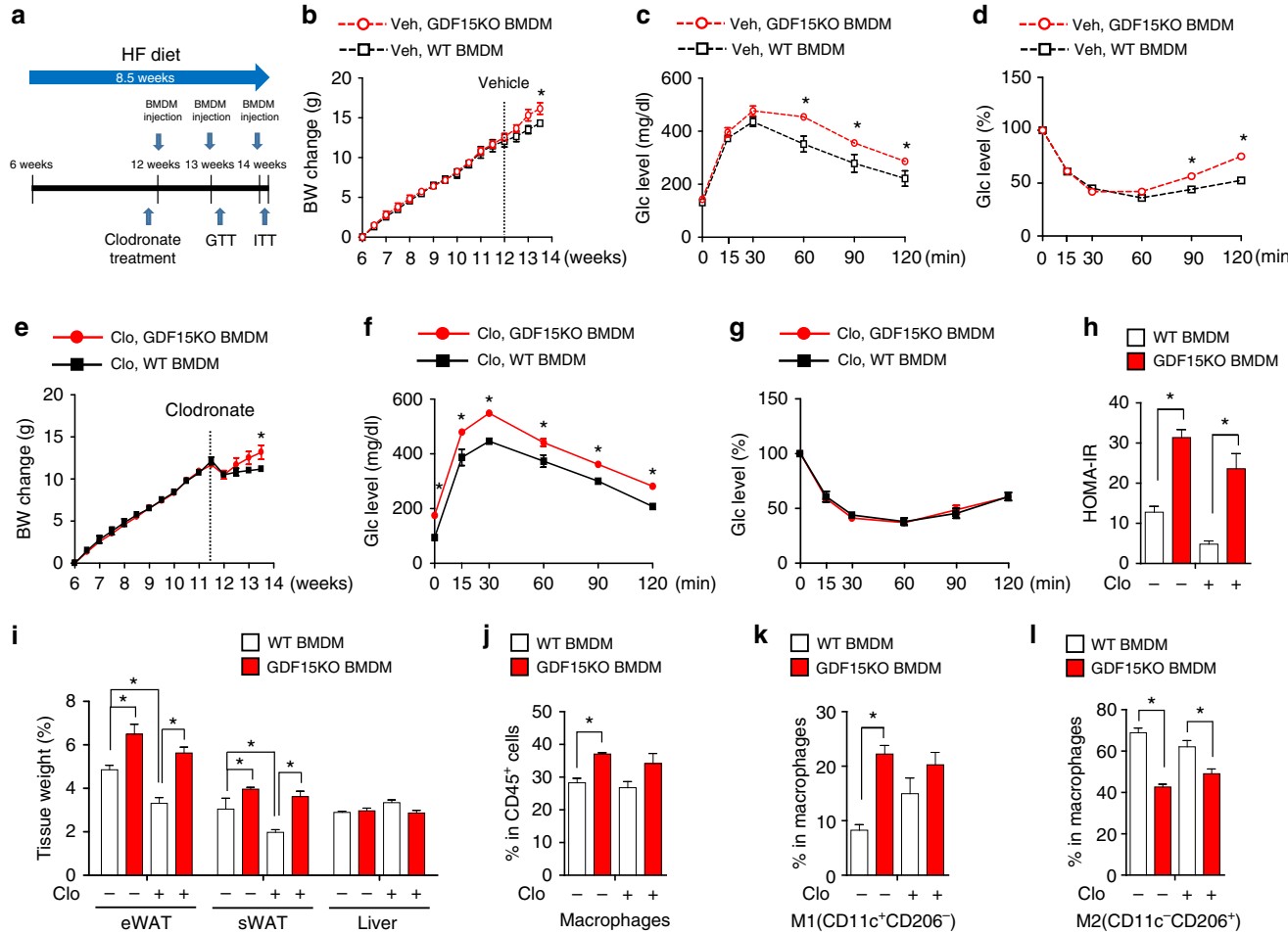

**Fig. 5** Adoptive transfer of GDF15KO macrophages aggravates glucose tolerance. **a** Diagram representing the experimental design. **b** Body weight change of PBS-liposome-treated HFD-fed mice intravenously injected with BMDMs from WT or *Gdf15*-KO ($n = 5$ per group). **c**, **d** IPGTT after a 16 h fast and an ITT after a 4 h fast ($n = 5$ per group). **e** Body weight change of clodronate liposome–treated HFD-fed mice injected with BMDMs from WT or *Gdf15*-KO ($n = 5$ per group). **f**, **g** IPGTT after a 16 h fast and ITT after a 4 h fast ($n = 5$ per group). **h** HOMA-IR of PBS or clodronate liposome-treated HFD-fed mice injected with BMDMs from WT or *Gdf15*-KO ($n = 5$ per group). **i** Tissue weight of PBS or clodronate liposome-treated HFD-fed mice injected with BMDM from WT or *Gdf15*-KO ($n = 5$ per group). **j**–**l** FACS analysis of total macrophages (CD45$^+$/F4/80$^+$/CD11b$^+$) and M1 (CD11c$^+$/CD206$^-$) and M2 (CD11c$^-$/CD206$^+$) macrophages infiltrated into the eWAT. Data represent means ± SEM and are representative of three independent experiments. *$p < 0.05$ (two-tailed Student's *t*-test)

in mouse BMDMs. Because the proinflammatory molecules inhibit PPARγ activity, whereas interleukin-4 (IL-4) stimulates PPARγ activity, in human and murine macrophages and DCs[24], we sought to identify factors that were expressed in murine macrophages in response to IL-4. Of the five significantly upregulated genes, *Gdf15* was effectively induced by IL-4 (Supplementary Fig. 4C, Supplementary data 1). Gene Ontology (GO) biological process analysis revealed that the upregulated and downregulated genes are involved in M2 macrophage activation by IL-4 (Supplementary Fig. 4D). This result is consistent with the role of IL-4 in macrophage activation and polarization[25]. A complete list of over-represented and under-represented processes is provided in Supplementary data 2 and 3. Next, we examined the induction of five autocrine and paracrine factors in vitro. As observed in human macrophage transcriptomes, in mouse cells rosiglitazone clearly induced expression of the five genes listed above, whereas recombinant IL-4 (rIL-4) induced only *Angptl4* and *Gdf15* (Fig. 3a).

ANGPTL4 is a matricellular protein that has been implicated in multiple inflammation-associated diseases[26]. IL-4 promotes induction of *Angptl4* by rosiglitazone in wild-type, but not STAT6-deficient, macrophages[24]. We were more interested in

GDF15 (also known as macrophage inhibitory cytokine 1), which is regarded as an autocrine molecule that inhibits LPS-induced TNF production by macrophages[27]. Although GDF15 has been implicated in metabolic disease and inflammation, the details of its roles in these processes remain unknown[28]. To better understand the expression pattern of *Gdf15*, we examined rosiglitazone- or rIL-4-treated mouse BMDMs, and found that BMDMs treated with rosiglitazone or rIL-4 expressed higher levels of *Gdf15* mRNA and protein than control cells (Fig. 3b, c).

STAT6, a major transcription factor regulated by the Th2 cytokines IL-4 and IL-13, also promotes the maturation of alternatively activated macrophages[10,29]. To verify the role of STAT6 in IL-4–mediated *Gdf15* expression, we prepared BMDMs from *Stat6*-knockout (KO) mice and examined *Gdf15* expression upon treatment with rIL-4 and rosiglitazone. BMDMs from *Stat6*-KO mice (Supplementary Fig. 4E) exhibited reduced basal expression of *Gdf15* (Fig. 3d). Consistent with a previous report[24], STAT6 deficiency abrogated rIL-4-stimulated expression of *Arg1*, *Fizz1* (found in inflammatory zone), and *Ym1*, which are markers of M2-like macrophages[30,31]. In addition, expression of *Arg1*, *Fizz1*, and *Ym1* in STAT6-deficient macrophages was not altered by rosiglitazone treatment (Fig. 3e–g). As expected, rIL-4- or

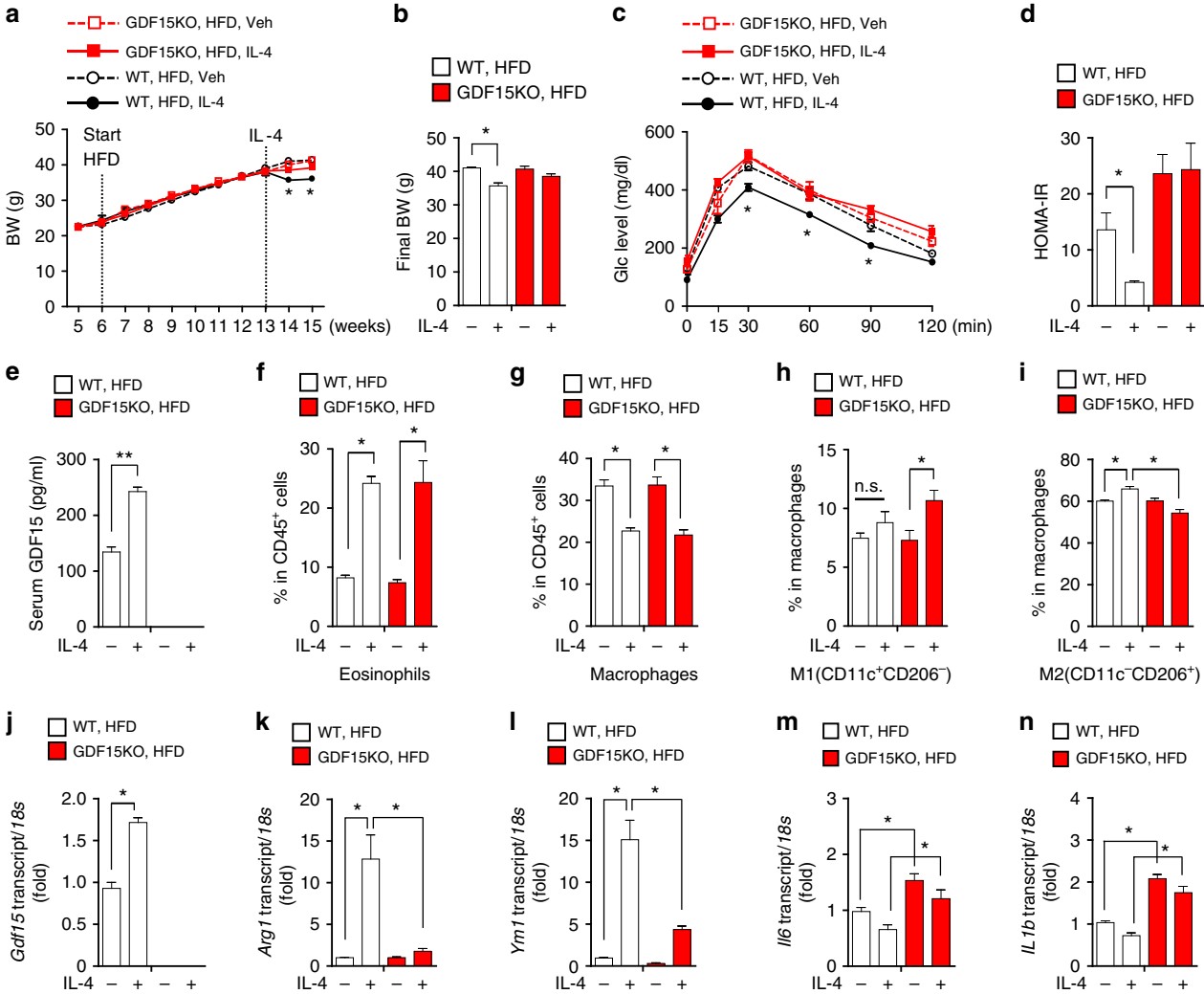

**Fig. 6** GDF15 deficiency prevents improvement of insulin sensitivity by rIL-4. **a, b** Body weight of WT and *Gdf15*-KO mice fed a HFD for 7 weeks and then intraperitoneally injected with rIL-4 (50 μg/kg) for 2 weeks (*n* = 5 per group). **c** WT or *Gdf15*-KO mice fed a HFD after injection of IL-4 were subjected to an IPGTT after a 16 h fast. **d** HOMA-IR of HFD-fed WT and *Gdf15*-KO mice after treatment with rIL-4. **e** Serum GDF15 level of HFD-fed WT and *Gdf15*-KO mice after treatment with rIL-4. **f–i** FACS analysis of eosinophils (CD45$^+$/CD11b$^+$/SiglecF$^+$), macrophages (CD45$^+$/F4/80$^+$/CD11b$^+$), and M1 (CD11c$^+$/CD206$^-$) and M2 (CD11c$^-$/CD206$^+$) macrophages infiltrated into eWAT. **j** Real-time PCR analysis of *Gdf15* mRNA in CD11b$^+$ cells from eWAT of WT and *Gdf15*-KO mice treated with rIL-4. **k–n** Real-time PCR analysis of *Arg1, Ym1, Il6*, and *Il1b* mRNAs in CD11b$^+$ cells from eWAT of WT and *Gdf15*-KO mice treated with rIL-4. Data are expressed as means ± SEM and are representative of three independent experiments. *$p < 0.05$ (two-tailed Student's *t*-test)

rosiglitazone-stimulated *Gdf15* expression was markedly attenuated in *Stat6*-KO BMDMs (Fig. 3h).

To obtain further support for the hypothesis that both PPARγ-mediated and IL-4-mediated STAT6 pathways govern *GDF15* expression in macrophages, we examined *GDF15* promoter activity using reporter constructs fused to the human GDF15 (*hGDF15*) promoter. RAW264.7 cells were transiently transfected with a long (*hGDF15* −1739/+70) construct (containing the PPARγ [PPRE] and STAT6 [STAT6-RE] binding sites) and a short (*hGDF15* −660/+70) construct, and then treated with rosiglitazone or rIL-4 for 24 h (Fig. 3i). The short construct had no transcriptional activity, whereas the long construct yielded a significant increase in luciferase activity in both rIL-4- and rosiglitazone-treated cells (Fig. 3j). This observation suggests that PPARγ and STAT6 play critical roles in IL-4- and rosiglitazone-dependent *GDF15* expression in macrophages (Supplementary Fig. 4F). Thus, expression of GDF15 is augmented by PPARγ and STAT6 activation, both of which increase oxidative function, a

metabolic characteristic of alternatively activated M2-like macrophages[24,32].

**GDF15 stimulates oxidative metabolism of macrophages**. Because expression of GDF15 increased in both rosiglitazone- and rIL-4-treated BMDMs, we next investigated whether GDF15 regulates oxidative metabolism in macrophages. Because GDF15 action is dependent on TGF-β RII[33,34], we measured mRNA expression of TGF-β RI receptor kinases (*Alk1–7*), which undergo hetero-oligomerization with type II serine/threonine kinase receptors to transduce signals, usually via SMAD proteins. BMDMs and RAW264.7 cells exhibited differential expression of ALK isoforms: the former expressed *Alk1*, *2*, and *5*, and the latter *Alk2*, *3*, and *5* (Fig. 4a, Supplementary Fig. 5). Recombinant GDF15 (rGDF15) stimulated phosphorylation of SMAD2 and 3 but not SMAD1, 5, or 9 (Fig. 4b, Supplementary Fig. 15). rGDF15-induced phosphorylation of SMAD2 and 3 was abolished by SB431542, a potent and selective inhibitor of TGF-β RI receptor kinases (ALK4, 5, and 7) (Fig. 4c, Supplementary

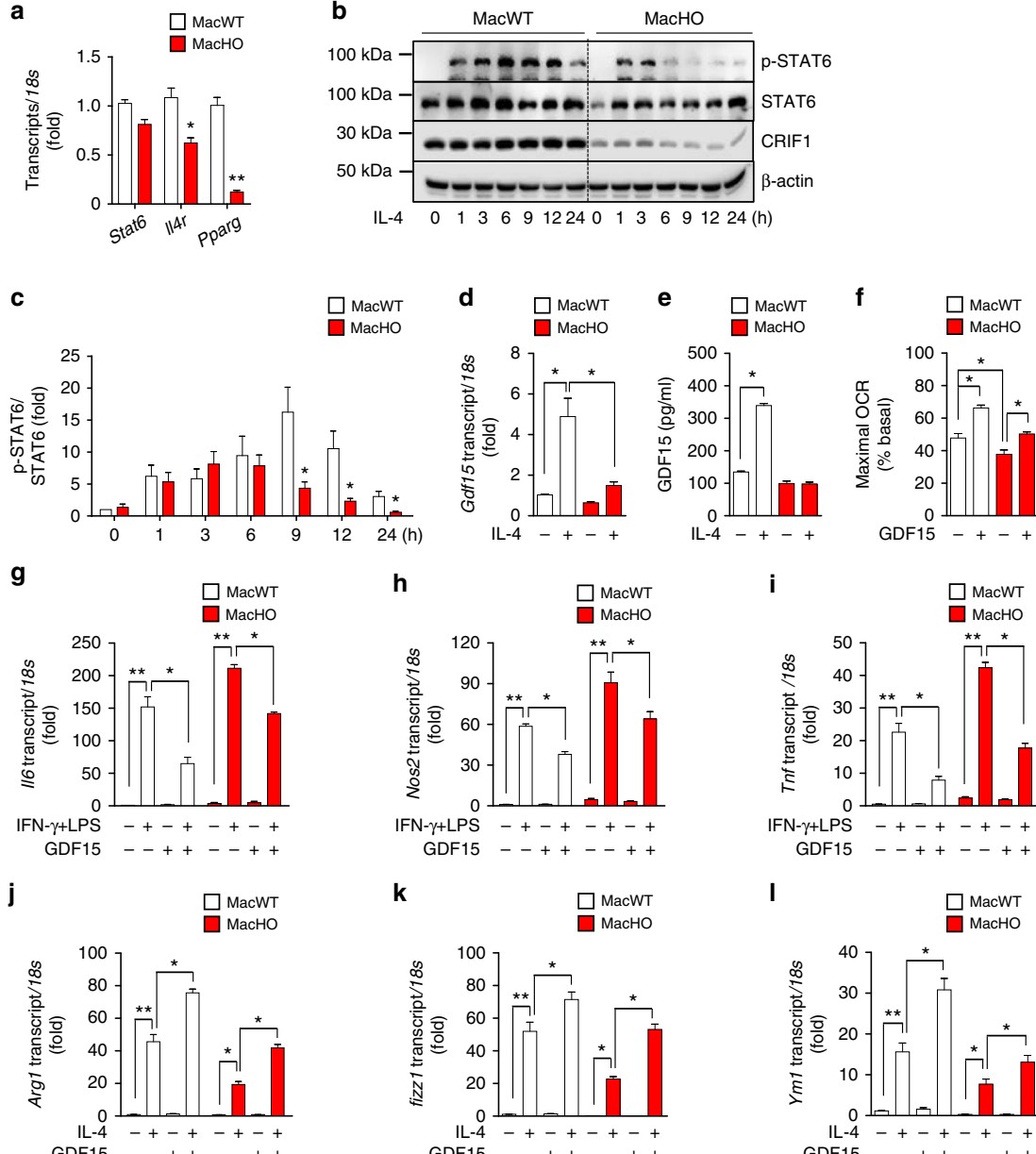

**Fig. 7** Macrophages from MacHO mice secrete less GDF15 upon stimulation with rIL-4. **a** Real-time PCR analysis of *Stat6*, *Il4r*, and *Pparg* gene expression in BMDMs from 8-week-old MacWT and MacHO mice. **b, c** Immunoblot analysis of STAT6 phosphorylation in BMDMs from 8-week-old MacWT and MacHO mice after treatment with rIL-4 (100 ng/ml). **d** Real-time PCR analysis of *Gdf15* mRNA expression in BMDMs from 8-week-old MacWT and MacHO mice in the presence/absence of rIL-4 (100 ng/ml). **e** GDF15 levels in culture supernatants of MacWT and MacHO BMDMs from 8-week-old mice after treatment with rIL-4 (100 ng/ml) for 48 h. **f** OCR in BMDMs from 8-week-old MacWT ($n = 5$) and MacHO mice ($n = 5$) treated with rGDF15 (300 ng/ml) for 24 h. **g–i** Real-time PCR analysis of *Il6*, *Nos2*, and *Tnf* mRNA after treatment with rGDF15 (300 ng/ml) in the presence or absence of IFN-γ (10 ng/ml) and LPS (100 ng/ml). **j–l** Real-time PCR analysis of *Arg1*, *Fizz1*, and *Ym1* mRNA expression after treatment with rGDF15 (300 ng/ml) in the presence/absence of rIL-4 (20 ng/ml). Data are expressed as means ± SEM and are representative of three independent experiments. *$p < 0.05$ or **$p < 0.01$ (two-tailed Student's *t*-test)

Fig. 16). These findings indicate that GDF15 transmits the SMAD2 and 3 activating signals via the TGF-β RI receptor kinases ALK4, 5, and 7.

Next, we investigated whether GDF15 regulates oxidative metabolism in macrophages[29]. rGDF15 significantly increased the maximal OCR; however, this increase was completely abolished by SB431542 (Fig. 4d, e). Thus, GDF15-mediated signaling increased oxidative function in macrophages by activating ALK4, 5, and 7. To confirm this finding, we analyzed FAO and its associated gene expression patterns in macrophages treated with rGDF15. Exposure of BMDMs to rGDF15 increased palmitate oxidation (Fig. 4f, g). Moreover, qRT-PCR revealed that GDF15-stimulated macrophages increased the expression of genes controlling fatty oxidation (*Cpt1*, *Acadm*, and *Acadvl*) (Fig. 4h). Furthermore, rGDF15 inhibited LPS- and IFN-γ–induced production of *Il6*, *Nos2*, and *Tnf* in macrophages, suggesting that GDF15 suppresses M1-like activation of macrophages (Fig. 4i–k). By contrast, rGDF15 augmented IL-4-mediated expression of *Arg1*, *Fizz1*, and *Ym1* in BMDMs (Fig. 4l–n).

**GDF15 deficiency in macrophages aggravates glucose tolerance.** To further characterize the role of macrophage GDF15 in systemic insulin resistance, we performed adoptive transfer of GDF15-deficient macrophages into macrophage-depleted mice[35]. To generate macrophage-depleted mice, clodronate liposome (clodronate) was injected intraperitoneally, resulting in approximately 70% depletion of eWAT macrophages after 24 h, in comparison with vehicle (PBS-liposome) injection (Supplementary Fig. 6A, 6B)[36–39]. The fluorescent dye PKH26, which mainly binds to the cell membrane, has been used as a macrophage tracer to locate transplanted cells in eWAT[40]. After intravenous injections of PKH26-stained BMDMs, the abundance of fluorescent macrophages was elevated in eWAT irrespective of clodronate treatment (Supplementary Fig. 6C–6E).

Next, to test whether GDF15 deficiency in macrophages affects systemic insulin resistance, we intravenously injected GDF15-deficient BMDMs ($1 \times 10^6$ per mouse) three times per week into mice on a HFD (Fig. 5a)[1]. Mice injected with *Gdf15*-KO BMDMs exhibited an increase in body weight regardless of clodronate treatment (Fig. 5b, e), which was largely due to an increase in WAT weight (Fig. 5i). Injection of GDF15-deficient BMDMs mildly aggravated the response to glucose and insulin challenges in the vehicle-treatment (Fig. 5c, d) and clodronate-treatment groups (Fig. 5f, g). In addition, injection of GDF15-deficient BMDMs induced an increase in homeostasis model assessment of insulin resistance (HOMA-IR) in both the vehicle-treatment and clodronate-treatment groups (Fig. 5h).

To further confirm that GDF15 deficiency affects the polarization of adipose macrophages, we analyzed the macrophage population in SVCs by fluorescence-activated cell sorting (FACS). The percentage of total macrophages (CD45$^+$/CD11b$^+$/F4/80$^+$) was elevated in the *Gdf15*-KO BMDM group (WT vs. *Gdf15*-KO: 28.25 ± 2.38% vs. 37.05 ± 0.64%, $p < 0.05$, Student's $t$-test) (Fig. 5j, Supplementary Fig. 7). The M1 population of macrophages was also increased in the *Gdf15*-KO BMDM group (WT vs. *Gdf15*-KO: 8.27 ± 1.75% vs. 22.23 ± 2.74%, $p < 0.05$, Student's $t$-test) (Fig. 5k). As expected, the M2 population of macrophages decreased in the *Gdf15*-KO BMDM group with or without clodronate treatment (WT vs. *Gdf15*-KO with PBS liposomes: 68.83 ± 3.88% vs. 42.60 ± 2.33%, $p < 0.05$, Student's $t$-test; WT vs. *Gdf15*-KO with clodronate: 62.00 ± 5.29% vs. 49.05 ± 4.00%; $p < 0.05$, Student's $t$-test) (Fig. 5l). The results of the adoptive transfer study suggest that GDF15 deficiency in macrophages is associated with exacerbation of glucose intolerance by an altered immune environment in white adipose tissue.

**GDF15 deficiency impedes IL-4-mediated metabolic recovery.** Given that GDF15 is required to promote oxidative metabolism in M2-like macrophages stimulated with IL-4, we investigated whether IL-4 regulates adipose inflammation and glucose metabolism in a GDF15-dependent manner. In these experiments, we intraperitoneally injected rIL-4 for 2 weeks into WT and *Gdf15*-KO mice, which were fed a HFD for 7 weeks. There was no difference in body weight between control and *Gdf15*-KO mice fed a HFD (Fig. 6a, b). Control WT mice treated with rIL-4 for 1 week exhibited the significant reduction in body weight, improvement of glucose tolerance, and reduction of plasma insulin levels, but these effects were absent in *Gdf15*-KO mice (Fig. 6a–d). Serum GDF15 levels were, as expected, only increased by IL-4 in WT mice (Fig. 6e). These observations suggest that the whole-body metabolic action of rIL-4 is mediated by GDF15.

The proportion of eosinophils in eWAT was significantly higher in both control and *Gdf15*-KO mice treated with rIL-4 (Veh vs. IL-4 treated: 7.19 ± 1.37% vs. 22.63 ± 4.30%, $p < 0.05$, Student's $t$-test) (Fig. 6f, Supplementary Fig. 8). The percentage of

macrophages in CD45$^+$ cells in eWAT was dramatically reduced following rIL-4 treatment (Veh vs. IL-4 treated: 35.76 ± 3.46% vs. 17.80 ± 5.36%, $p < 0.05$, Student's $t$-test) (Fig. 6g). The M1 population (CD11b$^+$, F4/80$^+$, CD11c$^+$) of macrophages was not increased upon rIL-4 treatment, but was more abundant in eWAT of *Gdf15*-KO mice following rIL-4 treatment (Fig. 6h). The M2 population (CD11b$^+$, F4/80$^+$, CD206$^+$) of macrophages increased upon rIL-4 treatment (Veh vs. IL-4 treated: 60.50 ± 1.05% vs. 65.83 ± 2.02%, $p < 0.05$, Student's $t$-test) (Fig. 6i). Interestingly, treatment with rIL-4 failed to increase the M2 population (CD11b$^+$, F4/80$^+$, CD206$^+$) in adipose tissue from *Gdf15*-KO mice (Veh vs. IL-4 treated: 60.07 ± 2.29% vs. 53.66 ± 3.05%, Student's $t$-test) (Fig. 6i).

To further characterize the role of GDF15 on adipose macrophage polarization, we analyzed the expression of polarization marker genes in CD11b$^+$ cells isolated from *Gdf15*-KO mice in response to rIL-4 administration. *Gdf15* and M2 activation-related genes such as *Arg1* and *Ym1* were strongly induced upon IL-4 treatment of CD11b$^+$ cells from WT mice treated with rIL-4, but the IL-4-mediated induction of those genes was markedly blunted in CD11b$^+$ cells from *Gdf15*-KO mice (Fig. 6j–l). M1 activation-related genes such as *Il6* and *Il1b* were upregulated in CD11b$^+$ cells from *Gdf15*-KO mice, but suppressed by rIL-4 treatment (Fig. 6m, n).

The improvement of glucose tolerance upon rIL-4 treatment in mice fed a HFD was absent in *Gdf15*-KO mice. Furthermore, GDF15 deficiency in macrophages resulted in a failure of M2 polarization in response to IL-4 stimulation. Hence, GDF15 maintains both the M2-polarization status in adipose tissue and systemic glucose homeostasis.

**GDF15 secretion is reduced in OxPhos-defective macrophages.** Given that macrophages from MacHO mice exhibited reduced STAT6-mediated responses to IL-4, we next examined the expression and secretion of GDF15, which is induced upon PPARγ and STAT6 activation. Expression of IL-4 receptor (*Il4r*) and *Pparg* mRNA in macrophages was significantly lower in MacHO than in MacWT, and *Stat6* was expressed at a lower level, although it was not significant (Fig. 7a). rIL-4 treatment caused robust induction of STAT6 phosphorylation in MacWT BMDMs (pTyr 641); however, MacHO BMDMs exhibited lower levels of STAT6 phosphorylation (Fig. 7b, c, Supplementary Fig. 17). Consistent with this, MacHO BMDMs expressed a lower level of IL-4 receptor than MacWT BMDMs, suggesting that these cells also expressed lower levels of GDF15[41]. Supporting this assumption, IL-4-stimulated expression and secretion of GDF15 were markedly reduced in MacHO BMDMs (Fig. 7d, e).

To ascertain whether GDF15 treatment increases the oxidative capacity of macrophages, we measured OCR in rGDF15-treated macrophages (Fig. 7f). rGDF15 increased maximal OCR in BMDMs from both MacWT and MacHO mice, indicating that GDF15 increases the oxidative function of macrophages. We next investigated whether rGDF15 suppresses M1-like activation and upregulates M2-like activation of MacHO BMDMs. MacHO macrophages pretreated with rGDF15 exhibited reduced expression of *Il6*, *Nos2*, and *Tnf* in response to IFN-γ and LPS treatment (Fig. 7g–i). Moreover, expression of IL-4-mediated M2 activation-related genes was restored by treatment with rGDF15 (Fig. 7j–l).

**rGDF15 reverses insulin resistance in MacHO mice.** We hypothesized that administration of rGDF15 would ameliorate systemic insulin resistance by restructuring the macrophage population in adipose tissue. To verify this, we treated MacHO mice exhibiting systemic insulin resistance and adipose infiltration by M1-like macrophages with rGDF15 (300 μg/kg) every

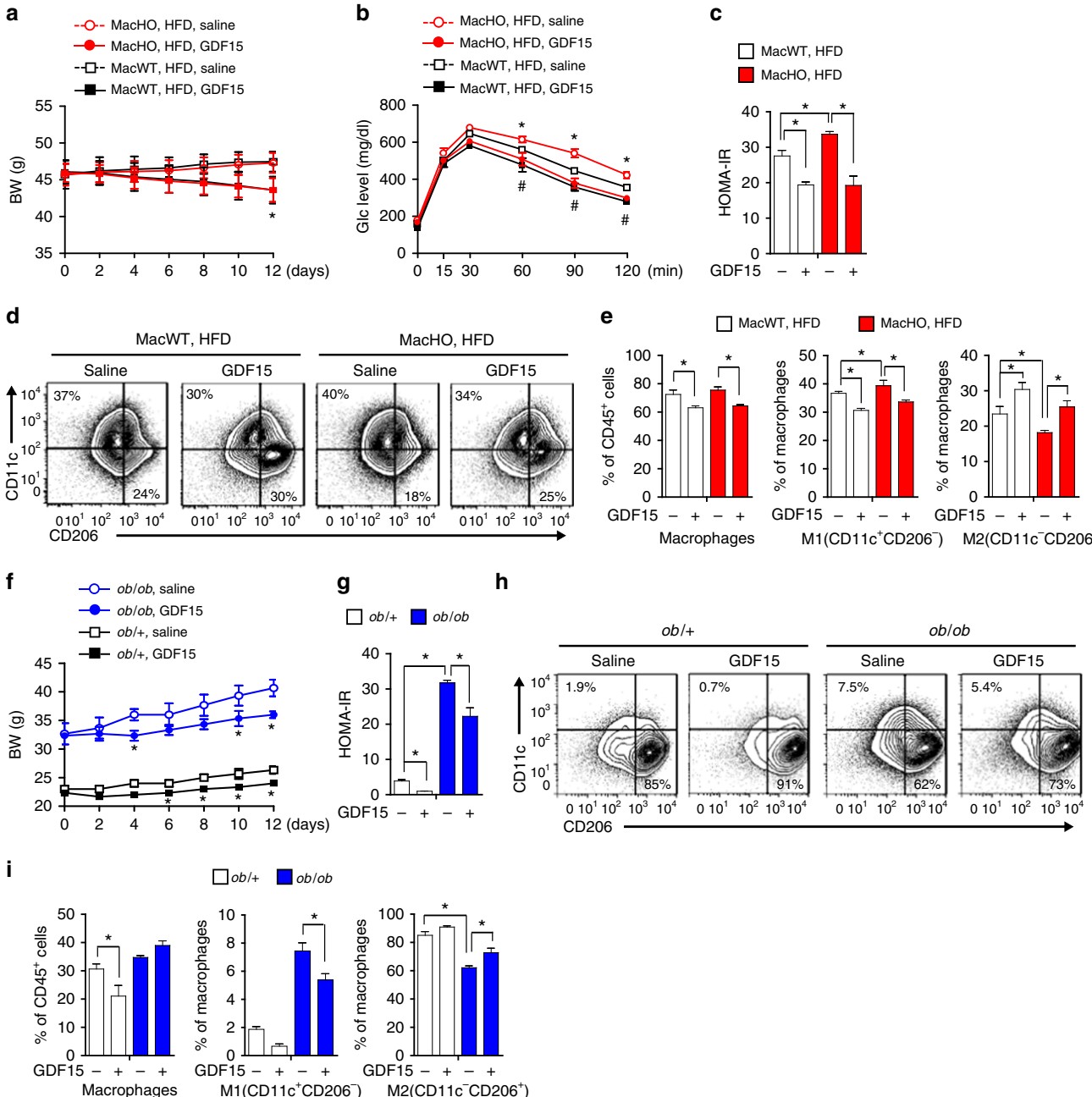

**Fig. 8** GDF15 reverses insulin resistance caused by reduced oxidative function. **a** Body weight changes in MacWT or MacHO mice fed a HFD for 12 weeks and intraperitoneally injected with rGDF15 (300 μg/kg) every other day for 2 weeks (n = 5 per group). **b** IPGTT performed after rGDF15 treatment. **c** HOMA-IR of HFD-fed MacWT and MacHO mice after treatment with rGDF15. **d**, **e** FACS analysis of macrophages (CD45$^+$/F4/80$^+$/CD11b$^+$) and M1 (CD11c$^+$/CD206$^-$) and M2 (CD11c$^-$/CD206$^+$) macrophages infiltrated into eWAT. **f** Body weight changes in ob/+ and ob/ob mice treated with rGDF15 (300 μg/kg) for 2 weeks (n = 5 per group). **g** HOMA-IR of ob/+ and ob/ob mice after GDF15 treatment. **h**, **i** FACS analysis of macrophages (CD45$^+$/F4/80$^+$/CD11b$^+$) and M1 (CD11c$^+$/CD206$^-$) and M2 (CD11c$^-$/CD206$^+$) macrophages infiltrated into eWAT of ob/+ and ob/ob mice. Data represent means ± SEM and are representative of three independent experiments. *p < 0.05 (two-tailed Student's t-test)

other day for 2 weeks. Systemic administration of rGDF15 to MacHO mice fed a HFD led to a slight reduction in body weight after 10 days, and improved both systemic glucose tolerance and insulin sensitivity (as demonstrated by a reduced HOMA-IR) (Fig. 8a–c, Supplementary Fig. 9A, 9B). Moreover, in comparison with vehicle, rGDF15 reduced the percentage of macrophages within the total CD45$^+$ cell population in SVCs of eWAT in both MacWT and MacHO mice (Fig. 8d, e). Notably, the proportion of M1-like macrophages within the total macrophage population in eWAT was reduced significantly, whereas that of M2-like

macrophages increased (MacWT: from 23.43 ± 3.72% to 30.40 ± 33.35%, p < 0.05; MacHO: from 18.20 ± 1.06% to 25.44 ± 3.01, p < 0.05, Student's t-test) (Fig. 8d, e, Supplementary Fig. 10). Thus, GDF15 improves glucose tolerance in MacHO mice fed a HFD by regulating macrophage polarity in adipose tissue.

**rGDF15 improves systemic glucose tolerance in ob/ob mice**. Finally, to determine whether GDF15 affects systemic glucose tolerance and macrophage infiltration in a rodent model of obesity, we treated ob/ob mice (in which M1-like macrophages

predominate in adipose tissue) with rGDF15. This treatment led to a reduction in weight, HOMA-IR, and improvement of systemic glucose tolerance and insulin sensitivity in both *ob*/+ and *ob*/*ob* mice (Fig. 8f, g, Supplementary Fig. 9C–G). The total macrophage population in SVCs of eWAT of *ob*/+ mice was reduced, as was the percentage of M1-like macrophages (Fig. 8h, i, Supplementary Fig.11). Consistent with this, the percentage of M2-like macrophages increased (*ob*/+: $85.33 \pm 4.95$ to $90.9 \pm 1.3\%$, $p < 0.05$; *ob*/*ob*: $62.1 \pm 1.90$ to $72.76 \pm 5.55\%$, $p < 0.05$, Student's *t*-test) (Fig. 8h, i). Thus, these findings suggest that GDF15 is a critical systemic hormonal regulator of both macrophage-induced adipose inflammation and systemic insulin resistance.

## Discussion

Following the seminal discovery of the contribution of ATMs to obesity-related metabolic diseases, these cells were extensively characterized with the goal of identifying potential therapeutic targets for insulin resistance associated with excessive adiposity[1]. The current evidence supports the idea that macrophages residing in lean adipose tissue are alternatively activated (M2-like), whereas those in obese adipose tissue are both markedly more abundant and predominantly classically activated (M1-like) phenotypes; the latter cells promote adipose inflammation and insulin resistance.

CRIF1, an essential mammalian mitoribosomal factor required for the synthesis and insertion of mitochondrial-encoded OxPhos polypeptides into the mitochondrial membrane, is abundantly expressed in skeletal muscle[20,42]. Loss of *Crif1* leads to OxPhos dysfunction due to both aberrant synthesis and defective insertion of OxPhos polypeptides into the inner mitochondrial membrane[20]. Consistent with this, we found that *Crif1* deficiency induced progressive mitochondrial OxPhos dysfunction in macrophages. In addition, *Crif1*-deficient macrophages exhibited typical features of a phenotypic switch toward M1-like cells. Moreover, *Crif1*-deficient macrophages exhibited reduced Th2 cytokine responses, as reflected by reductions in the expression levels of *Arg1*, *Ym1*, and *Gdf15* expression. Although impaired oxidative function was previously observed in M1-like macrophages, it remained unclear whether primary OxPhos deficiency in macrophages causes insulin resistance associated with adipose inflammation. Here, we showed that reduced mitochondrial oxidative function in macrophages precipitates adipose inflammation and systemic insulin resistance in mice fed a HFD. Pharmacological PPARγ agonists do not influence M2 marker expression in resting or M1 macrophages, but can prime native monocytes to express an M2 phenotype[22], and induction of an M2-like phenotype by PPARγ agonists is associated with elevated oxidative function. Increases in oxidative function induced by PPARγ agonists are not limited to macrophages, but are also observed in adipose tissue[43]. Therefore, potential therapeutic agents designed to attenuate M1-like macrophage polarization and improve systemic insulin resistance may, along with weight loss, be efficacious for obese diabetic patients.

STAT6 and PPARγ are the primary activators of M2 polarization of macrophages[4,11]. Activation of STAT6 and PPARγ increases oxidative metabolism, which is dependent on mitochondrial function, resulting in secretion of IL-10, TGF-β1, Ym1, and FIZZ1. These proteins may act in turn as local paracrine factors to control the microenvironment in adipose tissue. In line with these findings, paracrine factors possibly originating from stromal cells and adipocytes in adipose tissue may determine the polarization status of ATMs. In this study, we showed that GDF15 levels, which are transcriptionally co-regulated by STAT6 and PPARγ, are increased by Th2 cytokines and PPARγ agonists.

Our observations of elevated respiration, palmitate oxidation, and expression of FAO-related genes in GDF15-treated macrophages suggest that GDF15 shifts the metabolic phenotype of macrophages by increasing their oxidative function (Fig. 4). Reprogrammed energy metabolism, characterized by increased oxidative function, may be linked to inhibition of LPS- and IFN-γ–induced M1-like polarization, and may potentiate M2-like polarization stimulated by the Th2 cytokine IL-4[44]. The therapeutic potential of GDF15 has been tested in model animals with pathologic conditions characterized by abnormal macrophage accumulation. For example, elevated expression of GDF15 in macrophages protects ApoE-deficient mice from atherosclerosis[45]. In another study, GDF15 transgenic mice were resistant to diet- and genetically induced obesity and exhibited improved insulin sensitivity due to a reduction in NLRP3 inflammasome activity in adipose tissue, in the context of obesity and insulin resistance[46]. These observations corroborate our findings that GDF15 is a key player in the regulation of macrophage-mediated adipose inflammation.

GDF15 is a marker of mitochondrial stress in vitro and in vivo, and plasma GDF15 levels increase under mitochondrial dysfunction or disease[47–49]. The elevated plasma GDF15 levels in mice and humans with mitochondrial OxPhos dysfunction may result from higher secretion of GDF15 in affected tissues, including skeletal muscle[50]. However, induction and secretion of GDF15 in cells with compromised OxPhos function is largely dependent on cell type. Specialized cell types, such as pancreatic beta cells and macrophages, have a limited capacity to induce GDF15 secretion in response to OxPhos dysfunction, e.g., in the case of Crif1 deficiency[51]. Based on these observations, we speculate that decreased GDF15 secretion in Crif1-deficient macrophages is a cell type-specific response to mitochondrial stress, which is different from the response in other tissues.

It has recently been reported that GDF15 specifically binds to the GDNF family receptor a-like (GFRAL) with high affinity, and that GFRAL requires association with the co-receptor RET to elicit intracellular signaling in response to GDF15 stimulation in the hindbrain[52,54,55]. However, there are several reports that GDF15 action is dependent on the TGF-β RII, suggesting alternate signaling mechanisms via different receptor interactions which may be distinct in the hindbrain and the periphery[33,34]. We have also shown that rGDF15 stimulates the phosphorylation of SMAD2 and 3 in macrophages via the TGF-β1 receptor kinase, ALK4, 5, and 7. Thus, although Gfral has been identified as an important receptor for GDF15, GDF15 signaling through the TGF-β receptor in peripheral tissues and macrophages cannot be ignored. Although pharmacological administration of GDF15 in rodents and primates effectively decreases body weight, the physiological role of GDF15 in regulation of body weight remains unclear[53–55]. In this study, GDF15-deficient mice on HFD exhibited no difference in body weight in comparison with WT mice (Fig. 6a). The serum concentration of GDF15 in WT mice on HFD was around 120 pg/ml, higher than that of mice on NCD (65 pg/ml). Recent work showed that weight loss in mice is associated with very high plasma GDF15 concentration, 300–600 pg/ml[56–58]. These findings indicate that the weight loss effect of GDF15 is associated with pharmacologically elevated plasma concentrations of this protein.

In summary, the results of this study demonstrate that reduced macrophage oxidative function determines systemic insulin resistance and adipose inflammation. GDF15 is an active autocrine and paracrine hormone that regulates the immune microenvironment in adipose tissue. Hence, our findings suggest that it would be worthwhile to test the ability of GDF15 to improve systemic metabolic homeostasis and decrease obesity in patients.

## Methods

**Isolation of BMDMs and cell culture.** Mouse BMDMs were isolated from bone marrow cells prepared from femur and tibia and cultured at 37 °C/5% $CO_2$ for 7 days in DMEM supplemented with 10% FBS, 1% penicillin–streptomycin (GIBCO, Thermo Fisher, Waltham, MA, USA), and recombinant murine M-CSF (20 ng/ml; R&D Systems, Minneapolis, MN, USA). Cell purity, as determined by FACS analysis of CD11b and F4/80 expression, was >95%. The murine macrophage cell line RAW264.7 (ATCC) was cultured at 37 °C/5% $CO_2$ in RPMI supplemented with 10% FBS and 1% penicillin-streptomycin.

**Mice.** Floxed *Crif1* (*Crif1^f/f*) mice were generated as previously described[59]. *Gdf15*-KO mice were obtained from Dr. Se-Jin Lee at the Johns Hopkins University School of Medicine[60]. LysM-*Cre* transgenic and *Stat6*-KO mice were purchased from the Jackson Laboratory. *Crif1^f/f* mice were bred with LysM-*Cre* mice to generate MacHO mice. All mice, apart from *Stat6*-KO (BALB/c) mice, were of the C57BL/6 background. Eight-weeks-old male mice were used for all experiments except HFD fed mice that were noted in figure legends. A HFD comprising 60% fat was purchased from Research Diets Inc (D12492, New Brunswick, NJ, USA). Mice were maintained in a controlled environment (12 h light/12 h dark cycle; humidity, 50–60%; ambient temperature, $22 \pm 2$ °C) and fed a NCD (except as noted). Five-week-old male *ob/+* control mice and *ob/ob* C57BL/6 J *Lep(−/−)* mice were purchased from Envigo (Indianapolis, IN, USA). All experiments were performed in three independent replicates, using three or four mice per group. The experimental procedures were approved by the Institutional Animal Care and Use Committee of the Chungnam National University School of Medicine (Daejeon, Korea).

**Plasmid constructs.** The pGL3Basic-human GDF15 (−1739/+70) luciferase reporter construct was kindly provided by Dr. Y Moon (Pusan National University, Korea). The pGL3basic-GDF15 deletion (STAT RE [-660 bp]) plasmid was constructed by inserting the PCR-amplified fragment of the human *GDF15* promoter into the *KpnI/XhoI*-digested pGL3Basic vector (Promega Corp., Madison, WI, USA). The nucleotide sequence of the 5′-flanking region of the *GDF15* promoter was scanned, and one putative STAT6 binding site (-670 to -661, TTCCTGGAA) and one PPRE binding site (−1169 to −1156, AGTGCAATGGTGA) were identified. The nucleotide sequences of all plasmids were confirmed by automated sequencing.

**Immunoblot analysis.** Aliquots of cell lysate (50 μg/lane) were separated by electrophoresis on 10% or 12% SDS polyacrylamide gels. Anti-phospho-STAT6, anti-STAT6, anti-phospho-SMAD2 (#3104), anti-SMAD2, anti-phospho-SMAD3, anti-SMAD3, anti-phospho-SMAD1, 5, and 9, anti-SMAD1, 5, and 9, anti-phospho-ERK1/2 (44/42 MAPK) (#4370), anti-ERK1/2 (#9102), anti-phospho-p38, anti-p38, anti-phospho-JNK, anti-JNK, and anti-SDHA (#5893) antibodies were purchased from Cell Signaling Technology (Beverley, MA, USA). Anti-NDUFA9 and COX-4 were purchased from Santa Cruz Biotechnology (Dallas, TX, USA). Anti-UQCRC2 (ab14745) was purchased from Abcam (Cambridge, UK). Anti-ATP5A1 was purchased from Invitrogen (Thermo Fisher). Secondary antibodies (goat anti-mouse and goat anti-rabbit) were obtained from Santa Cruz Biotechnology. Images were scanned on an ODYSSEY instrument and quantified using Image Studio Digits (LI-COR Biosciences, Lincoln, NE, USA). Full length uncropped blots are presented in Supplementary Figures 12–17.

**RNA isolation and quantitative real-time PCR.** Total RNA was isolated using TRIzol (Life Technologies, Thermo Fisher). Complementary DNA (cDNA) was prepared from total RNA using M-MLV Reverse Transcriptase and oligo-dT primers (Invitrogen). Quantitative real-time PCR was performed using cDNA, QuantiTect SYBR Green PCR Master Mix (QIAGEN, Hilden, Germany), and specific primers. Primers used in this study are described in Supplementary Table 1. Relative expression was calculated by normalizing against 18S ribosomal RNA on a 7500 Real-Time PCR System Software (v2.0.6, Applied Biosystems, Thermo Fisher).

**Reagents.** SB431542, PD98059, SB203580, SB202190, oligomycin, carbonyl cyanide m-chlorophenyl hydrazone (CCCP) were from Sigma-Aldrich (St. Louis, MO, USA). Recombinant human GDF15 containing an N-terminal 6-His tag was purchased from R&D Systems (#957-GD/CF). Lyophilized protein was reconstituted with sterile 4 mM HCl and stored at −70 °C.

**OCR and FAO.** OCR was measured using a Seahorse XF-24 analyzer (Seahorse Bioscience Inc., North Billerica, MA, USA). Briefly, BMDMs were seeded in XF-24 plates ($4 \times 10^4$ cells in 200 μl of growth medium per well) and placed in a 37 °C/5% $CO_2$ incubator for 24 h. The sensor cartridge was placed into the calibration buffer supplied by Seahorse Bioscience and incubated at 37 °C in a non-$CO_2$ incubator for 4 h before the experiment. Immediately before measurement, the cells were washed and cultured (at 37 °C in a non-$CO_2$ incubator) in glucose-free DMEM medium lacking sodium bicarbonate. The following compounds were added depending on the experiment: glucose (25 mM); oligomycin A (2 μg/ml); CCCP (optimized

concentration; 5 μM), rotenone (2 μM), or palmitate-BSA (2 mM). OCR was automatically recorded by the sensor cartridge and calculated using the Seahorse XF-24 software.

**Biochemical measurement of serum components.** Blood was collected from the heart under general anesthesia and centrifuged at 800×*g* for 5 min, and the supernatant was collected. Serum insulin (Alpco Diagnostics, NH, USA) and GDF15 (R&D Systems) levels were measured using ELISA kits. Biochemical measurement of free fatty acid and total cholesterol was performed on a Hitachi 7180 auto analyzer using Wako reagents (Wako Pure Chemical Industries, Osaka, Japan).

**Isolation of SVCs.** To purify SVCs, fat pads were excised from male mice and minced in collagenase buffer (135 mM NaCl, 4.7 mM KCl, 2.5 mM $CaCl_2$, 1.25 mM $MgSO_4$, 10 mM HEPES, 3.5% BSA, 0.01% glucose, and 1 mg/ml collagenase type I [Worthington Biochemical, Lakewood, NJ, USA]). The minced tissues were digested at 37 °C for 40 min with gentle shaking, and then filtered through a 100 μm filter. Digested cells were obtained by centrifugation at 700×*g* for 5 min and incubated with RBC lysis buffer. The remaining cells were washed twice with PBS.

**Flow cytometry analysis of SVCs.** To identify M1 and M2 macrophages and eosinophils, SVCs were stained with the Live/Dead marker 7AAD (BD Bioscience, Thermo Fisher), followed by anti-CD45, anti-CD11b, anti-F4/80, and anti-SiglecF antibodies. To detect intracellular CD206, stained cells were fixed and permeabilized with Cytofix/Cytoperm (BD Biosciences), and then stained with anti-CD206 antibody. To identify CD4 T cells, CD8 T cells, and B cells, SVCs were stained with the Live/Dead marker 7AAD, followed by anti-CD45, anti-CD3, anti-CD4, anti-CD8, and anti-B220 antibodies (all from BioLegend, San Diego, CA, USA). The antibodies used are listed in Supplementary Table 2. All stained cells were examined on a FACS Canto II flow cytometer (BD Biosciences), and data were analyzed using the FlowJo software (FlowJo, LLC, Ashland, OR, USA).

**Flow cytometry analysis of blood cells.** To identify monocytes and granulocytes, mouse blood was incubated for 5 min with RBC lysis buffer and washed twice with PBS. The remaining cells were stained with the Live/Dead marker 7AAD, followed by anti-CD45, anti-CD 11b, anti-Ly6G, and anti-Ly6C antibodies (see Supplementary Table 2).

**Flow cytometry of bone marrow cells.** Bone marrow cells were incubated with RBC lysis buffer for 5 min, and then washed twice with PBS. The remaining cells were stained with the Live/Dead marker 7AAD, followed by anti-CD45, anti-CD11b, anti-Ly6G, anti-Ly6C, anti-CD3, anti-CD4, anti-CD8, and anti-B220 antibodies (see Supplementary Table 2).

**Purification of CD11b$^+$ cells from SVCs.** To obtain the CD11b$^+$ cell fraction, SVCs were enriched by magnetic¯activated cell sorting (MACS) using CD11b MicroBeads (Miltenyi Biotec, Bergisch Gladbach, Germany). The enriched CD11b$^+$ cells were then stained with anti-CD45, anti-CD11b, and anti-F4/80 antibody. Staining was confirmed on a FACS Canto II flow cytometer, and the cells were then processed for qRT-PCR.

**Macrophage depletion and reintroduction with PKH26 labeling.** For adoptive transfer in Fig. 5, clodronate and PBS liposomes (F70101C-AC) were purchased from FormuMax (Sunnyvale, CA, USA). The mice were intraperitoneally injected with clodronate and PBS liposomes (100 μl/mice) and fed a HFD. After 2 days of treatment with clodronate or PBS liposomes, BMDMs ($1 \times 10^6$) were injected into the tail vein of each group of mice once weekly for 3 weeks.

For supplementary Figure 6, PKH26 linker (P9691, Sigma, St. Louis, MO, USA) was used to track the migration of BMDMs. BMDMs ($1 \times 10^6$) for injection were stained with PKH linker according to the manufacturer's instructions. Two days after second injection, the ATMs were immediately isolated from eWAT and analyzed on a FACS Canto II flow cytometer.

**Administration of rIl-4 and rGDF15.** For experiments using rIL-4, C57BL/6 mice were fed a HFD and given intraperitoneal injections of rIL-4 (2 μg/mouse, 214-14, PeproTech, Rocky Hill, NJ, USA) or PBS alone every day for 2 weeks. For experiments using rGDF15, MacWT, MacHO, *ob/+* and *ob/ob* mice were given intraperitoneal injections of rGDF15 (8 μg/mouse, 957-GD/CF, R&D Systems) or 4 mM HCl alone every other day for 2 weeks.

**Histological analysis.** Tissue samples were obtained from 8-week-old male MacWT and MacHO mice, 16-week NCD-fed mice or 10-week HFD-fed MacWT and MacHO mice (*n* = 3 per each group). Samples for light microscopy were fixed in 4% paraformaldehyde for 4 h. Paraffin embedding, sectioning, H&E and Masson's trichrome staining, and immunohistochemistry (IHC) were performed according to standard protocols. For IHC, the tissue sections were incubated with primary antibodies (anti-F4/80; 1:100; Abcam) for 16 h at 4 °C, and binding was

detected using the Polink-1 HRP Rat-NM DAB Detection System (GBI Labs, Bothell, WA, USA).

**Intraperitoneal glucose and insulin tolerance tests**. For the intraperitoneal glucose tolerance test (IPGTT), mice were fasted for 16 h before glucose (2 g/kg of body weight) was injected into the intraperitoneal cavity. Blood glucose levels were measured 0, 15, 30, 60, and 90 min later using a glucometer (ACCU-CHEK, Roche Diagnostics Corporation, Indianapolis, IN, USA). The insulin tolerance test (ITT) was performed by measuring blood glucose after a 6 h fast followed by intraperitoneal injection of insulin (0.75 U/ kg body weight; Humalog, Lilly, USA).

**Microarray analysis of cDNA**. Total RNA was prepared from BMDMs isolated from 8-week-old male $Crif1^{f/f}$ mice. RNA amplification and labeling were performed using Cyanine 3-labeled cRNA (complementary RNA) was generated from Agilent's Low RNA Input Linear Amplification kit with 500 ng total RNA (Agilent Technologies, Santa Clara, CA, USA). Labeled cRNA was applied microarray (Agilent technologies, 8 × 60 K) using Agilent's Gene Expression Hybridization Kit (Agilent Technologies). Data were extracted on a G2565CA DNA Microarray Scanner using the Agilent Feature Extraction Software (Agilent Technologies) with default settings.

**Bioinformatics analyses**. All raw transcriptomic data are publicly available on Gene Expression Omnibus (GEO; www.ncbi.nlm.nih.gov/geo) under accession number GSE25088[24], as well as on GeneNetwork (www.genenetwork.org). Analysis of rosiglitazone-treated human macrophages (GSE25088) was performed using a customized gene set containing known humoral factors. Heat maps were built using GENE-E (Broad Institute, www.broadinstitute.org/cancer/software/GENE-E/).

**Statistical analyses**. Statistical analyses were performed using Prism 5 (GraphPad Software, La Jolla, CA, USA). Data are expressed as means ± standard error of the mean (SEM). All data from animal studies were analyzed by two-tailed Student's t-test. A p-value <0.05 was considered statistically significant.

**Data availability**. Microarray data that support the findings of this study have been deposited in the NCBI Gene Expression Omnibus (GEO) database with the primary accession code GSE110771. The other data that support the findings of this study are available from the corresponding author upon reasonable request.

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

## Acknowledgements

This work was supported by the Basic Science Research Program through the National Research Foundation of Korea (NRF), Ministry of Science, ICT, and Future Planning, Korea (NRF-2015R1A2A1A13000951, GRL 2017K1A1A2013124); the EPFL; the AgingX program of the Swiss Initiative for Systems Biology (51RTP0-151019 and 2013/153); and the NIH (R01AG043930). S.-B.J. was supported by a NRF grant from the Ministry of Education, Korea (NRF-2016R1C1B1016547). J.H.L. was supported by the NRF (NRF-2014R1A1A1006176), Korea.

## Author contributions

S.-B.J. performed most experiments, analyzed the data, and wrote the manuscript. M.S. conceived the study, designed experimental strategies, and wrote the manuscript. M.J.C., S.G.K., and S.E.L. performed experiments and analyzed data. J.Y.C. helped to write the manuscript. H.K.C., H.-S.Y., J.H.L., K.S.K., H.J.K., C.-S.K., R.W.W., and H.K. provided conceptual input. Y.K.K. and C.-H.L. provided the mice. D.R. and J.A. performed the bioinformatics analysis and edited the manuscript. H.K.L. helped with flow cytometry design and analysis.

## Additional information

**Competing interests:** The authors declare no competing interests.

