## [Peer Review File · Nature Communications]

Reviewers' comments:

Reviewer #1 (Remarks to the Author):

Summary

This manuscript by Shong and colleagues describes their finding that reduced mitochondrial oxidative function in macrophages precipitates adipose inflammation and systemic insulin resistance in obese mice. Their studies begin with CR6-interacting protein 1 (Crif1), in which myeloid-specific deletion shows macrophages with increased M1-like phenotype and the mice show insulin resistance and adipose tissue inflammation. These mice led the authors to identify growth differentiation factor 15 (GDF15) as a mediator of oxidative function in macrophages. Furthermore, they show that GDF15 is upregulated by IL4 and PPARgamma agonists and that its function downstream of IL4 is required for the metabolic benefits of IL4. The studies are very thorough and convincing and the conclusions drawn are supported by the data presented. I have only a few major and minor comments that should be addressed prior to acceptance.

Major comments

1. In order to address the role of macrophage GDF15, adoptive transfer of GDF15-deficient macrophage into clodronate treated mice was performed. However, tracing with PKH26 staining shows that very small proportion of macrophages are originating from transferred cells (Fig. S5). Does this data correlate with findings presented in Fig. 5a? Were macrophages injected once into mice? Is this sufficient? Are there other papers that support the use of this adoptive transfer technique?

Similarly, the authors show that GDF15 deficiency affects polarization of adipose tissue macrophages by using the model of GDF15-deficient macrophage injection. I don't think this experiment differentiates whether the changes in the M1 and M2 population originates from transferred vs. its own cells. This should be discussed and the interpretation of the data should be tempered.

2. In figure 8, the authors show that systemic administration of rGDF15 led to a significant reduction of body weight in MacHO and ob/ob mice. In contrast, GDF15 KO mice under HFD showed no difference in body weight compared to WT mice (Fig. 6a). What is the explanation of this discrepancy?

As the authors already discussed, recent studies suggest that GDF15-GFRAL axis is a novel regulator of metabolic homeostasis, primarily affecting food intake and resulting in body weight changes. Although these effects are mainly mediated by receptors expressed in the brain, I agree with the authors that different mechanisms might be responsible for its action in the periphery. Because the authors used global GDF15 KO mice rather than using macrophage-specific KO mice, it is possible there are additional defects in these animals. For instance, did the authors check food intake in their experiments?

3. In some experiments, HOMA-IR was presented as an index of insulin resistance (e.g. Fig. 8). Because HOMA-IR is a less informative index for measuring the degree of insulin resistance, I wonder if the authors tested ITT in these models.

Minor comments

1. Line 113: "reduction in reduced basal" -> "reduced basal"

2. Figure 2(a) legend: It is written that HFD was administered for 8 weeks, whereas the figure shows the data for more than 10 weeks after starting HFD. Please make clear for the duration of HFD.

3. Figure 3(a, e-h) legend: Please indicate the duration of IL-4/RSG treatment.

Reviewer #2 (Remarks to the Author):

Review for the manuscript entitled 'Reduced oxidative capacity in macrophages results in systemic insulin resistance and its reversal with GDF15 treatment by Jung et al.

In this study, Jung and colleagues generate mice with a myeloid cell-specific deletion of CR6-interacting factor 1 (Crif1) and demonstrate that these mice exhibit reduced mitochondrial oxidative phosphorylation, in addition to a shift towards a M1-like polarization of macrophages, whole-body insulin resistance and local adipose tissue inflammation ie. overall an unhealthy metabolic phenotype due to the deletion of Crif1 from macrophages. The authors further report that GDF15 expression is reduced in macrophages that harbor this Crif1 deletion, and that this is concomitant with impaired mitochondrial oxidation capacity. Conversely, treatment with the TZD rosiglitazone, and interleukin-4 (IL-4), appeared to reverse this phenotype and enhance macrophage GDF15 expression. Finally, the authors show that the macrophage GDF15 up-regulation (as stimulated by rosiglitazone and IL-4), enhances the oxidative capacity of the cell, causing it to switch from an M1-like state, to an M2-like state of polarization; this then reverses insulin resistance in HFD-fed mice and ob/ob mice that are null for Crif1 in macrophages. The authors conclude that reduced oxidative capacity specifically in macrophages is the key determinant of adipose tissue inflammation and whole-body insulin resistance, and that this can essentially be reversed by means of enhances GDF15 levels in these macrophages.

Overall, this is a good study that is very technically sound and contributes to the current literature on GDF15. However, there are major concerns regarding the novelty of the study and the fact that an almost identical mechanism and study has previously been published by the authors in 2016, albeit in a different cell type, skeletal-muscle cells.

1. The authors show that macrophage GDF15 levels are reduced with impaired mitochondrial oxidative function. However, in the literature, GDF15 has been shown to function as a 'mitochondrial stress marker', and levels are in fact increased under conditions of mitochondrial dysfunction and mitochondrial disease. For instance, this has been shown by numerous studies: (1) Fujita Y et al. (Secreted growth differentiation factor 15 as a potential biomarker for mitochondrial dysfunctions in aging and age-related disorder. *Geriatr Gerontol Int* 2016), (2) Montero R et al. (GDF-15 is elevated in children with mitochondrial disease and is induced by mitochondrial dysfunction. *PLoS One* 2016), and (3) Koene S et al. (Serum GDF15 levels correlate to mitochondrial disease severity and myocardial strain, but not disease progression in adult m.3242A>G carriers. *JIMD Rep* 2015). As such, how do the authors explain this discrepancy with the literature? High GDF15 levels have been related to mitochondrial dysfunction in several publications. Here the authors state that high GDF15 levels improve mitochondrial function, and that GDF15 is required for oxidative metabolism of the M2-macrophage. The authors also state (in the Introduction), that GDF15 is a macrophage-regulating autocrine and paracrine signaling factor that promotes anti-inflammatory responses in white adipose tissue. This discrepancy with the literature is concerning and should be addressed.

2. The authors have already published what appears to be a parallel study with an identical mechanism of GDF15 already in 2016 in *JBC* (Chung HK: Growth differentiation factor 15 is a myomitokine governing systemic energy homeostasis). Here, the authors deleted Crif1 from the muscle and showed an identical mechanism, whereby elevated GDF15 from muscle improved insulin sensitivity and oxidative metabolism in ob/ob mice. The novelty of this current study is therefore greatly reduced, as the authors have already published this mechanism and phenotype, albeit simply performed the study in a different cell type.

3. In Figure 4f, the OCR values should be plotted as absolute values, not baselined to palmitate, as

this can exaggerate the differences observed. The authors should use the absolute OCR values (pmol/min), as they have done for their other oxidative respiration graphs throughout the manuscript.

Reviewer #1 (Remarks to the Author):

Summary

This manuscript by Shong and colleagues describes their finding that reduced mitochondrial oxidative function in macrophages precipitates adipose inflammation and systemic insulin resistance in obese mice. Their studies begin with CR6-interacting protein 1 (Crif1), in which myeloid-specific deletion shows macrophages with increased M1-like phenotype and the mice show insulin resistance and adipose tissue inflammation. These mice led the authors to identify growth differentiation factor 15 (GDF15) as a mediator of oxidative function in macrophages. Furthermore, they show that GDF15 is upregulated by IL4 and PPARgamma agonists and that its function downstream of IL4 is required for the metabolic benefits of IL4. The studies are very thorough and convincing and the conclusions drawn are supported by the data presented. I have only a few major and minor comments that should be addressed prior to acceptance.

Major comments

Q1) In order to address the role of macrophage GDF15, adoptive transfer of GDF15-deficient macrophage into clodronate treated mice was performed. However, tracing with PKH26 staining shows that very small proportion of macrophages are originating from transferred cells (Fig. S5). Does this data correlate with findings presented in Fig. 5a? Were macrophages injected once into mice? Is this sufficient? Are there other papers that support the use of this adoptive transfer technique? Similarly, the authors show that GDF15 deficiency affects polarization of adipose tissue macrophages by using the model of GDF15-deficient macrophage injection. I don't think this experiment differentiates whether the changes in the M1 and M2 population originates from transferred vs. its own cells. This should be discussed and the interpretation of the data should be tempered.

A1) We thank the reviewer for pointing this out. We used modified adoptive transfer methods, which were described in previous studies (PMID: 22190646, PMID: 28192375, PMID: 26786981, PMID: 24466141, PMID: 27820698, PMID: 22745325, PMID: 15896292). Specifically, we applied the methods described in PMID: 22871793 and PMID: 22190646 for macrophage preparation, *in vitro* labeling, *in vivo* migration, and FACS analysis, which revealed elevated macrophage migration into adipose tissue in obese mice. In this study, the proportion of PKH+ cells in adipose tissue was about 2.1%. One group reported that the proportion of PKH+ cells in epididymal adipose tissue was about 5% following injection of PKH-stained mononuclear cells (PMID: 22190646). Another group reported that the proportions of CD19 + / PKH + cells were 1.3% and 2.7% in the visceral adipose tissue of lean and obese mice, respectively, following injection of PKH-stained B cells (PMID: 28192375). We rewrote the part of the manuscript describing the detailed method (Methods section of the revised manuscript, page 27, lines 585-595), and included relevant references (Reference number: 35-40).

A1-1) Does this data correlate with findings presented in Fig. 5a?

The experiments in Fig. 5 and supplemental Fig. 5 were performed in the same manner and applied a technique that is commonly used for adoptive transfer of macrophages (PMID: 22871793). As described in the "Methods" section, after 2 days of treatment with clodronate or PBS liposomes, BMDMs (1×10^6) were injected via the tail vein into each group of mice once weekly for 3 weeks.

For supplemental figure 5, PKH26 linker (P9691, Sigma, St. Louis, MO, USA) was used to track the migration of BMDMs. The BMDMs (1×10^6 cells) used for injection were stained according to the manufacturer's instructions. Two days after the second injection, the ATMs were immediately isolated from eWAT and analyzed on a FACS Canto II flow cytometer. As mentioned above, the proportion of PKH+ cells in epididymal adipose tissue of mice injected with BMDM was 2.1%.

In data displayed in Fig. 5, injection of macrophages into mice on a high-fat diet (HFD) was performed three times before staining to further increase macrophage migration in adipose tissue. In mice, HFD increases macrophage infiltration into adipose tissue (PMID: 14679176, PMID: 22190646). In sum, we believe that the data presented in Fig. 5a correlate with those in supplementary Fig. 5.

A1-2) Were macrophages injected once into mice? Is this sufficient?

To maximize the migration of adopted macrophages into adipose tissue, we performed the

injections twice (We have corrected the number of injections in Supplemental figure. 5 legend). We found more PKH+ macrophages in adipose tissue when we injected twice than when we injected only once (1.1% after the first injection versus 2.1% after the second injection). We modified the descriptions in the Method (Methods section of the revised manuscript, page 27, lines 586-595) section to provide detailed information about the adoptive transfer method. In addition, we added a reference regarding this method (PMID: 28192375, reference number: 37).

A1-3) Are there other papers that support the use of this adoptive transfer technique? Similarly, the authors show that GDF15 deficiency affects polarization of adipose tissue macrophages by using the model of GDF15-deficient macrophage injection. I don't think this experiment differentiates whether the changes in the M1 and M2 population originates from transferred vs. its own cells.

As mentioned above, we followed the method described in previous studies in which adoptive transfer of macrophages was performed in the context of metabolic and inflammatory diseases (PMID: 22190646, PMID: 28192375, PMID: 26786981, PMID: 24466141, PMID: 27820698, PMID: 22745325, PMID: 15896292). After injection into recipient mice, the fluorescently labeled monocytes are cleared from the circulation within several hours, and subsequently can be readily detected in adipose tissue, liver, and spleen. In lean animals, adipose tissue accumulation peaks at 1–2 days, remains stable for ~1 week, and gradually declines over the next 2 weeks (PMID: 22190646). In obese recipients, the increase in labeled ATMs is much larger, peaking at 2 days, with levels remaining relatively constant for at least 2–3 weeks.

We agree with the reviewer's opinion that it is difficult to determine whether the changes in the M1 and M2 populations were due to transferred cells or the mouse's own cells. However, the M1 population of macrophages was also significantly larger in the Gdf15-KO BMDM transfer group than in the control group (WT vs. Gdf15-KO: $8.27 \pm 1.75\%$ vs. $22.23 \pm 2.74\%$, $p < 0.05$). In addition, the M2 population of macrophages was also significantly smaller in the Gdf15-KO BMDM transfer group (WT vs. Gdf15-KO with clodronate: $62.00 \pm 5.29\%$ vs. $49.05 \pm 4.00\%$; $p < 0.05$). Therefore, we suggested that GDF15 deficiency in macrophages exacerbates glucose intolerance by altering the environment through adoptive transfer in WAT. As the reviewer recommended, we tempered the interpretation of the data in the Results section of the revised manuscript (page 13, lines 280-282).

Q2) In figure 8, the authors show that systemic administration of rGDF15 led to a significant reduction of body weight in MachO and ob/ob mice. In contrast, GDF15 KO mice under HFD showed no difference in body weight compared to WT mice (Fig. 6a). What is the explanation of this discrepancy?

A2) We thank the reviewer for pointing this out. We measured the serum concentration of GDF15 using the GDF15 ELISA kit (R&D Systems) in WT mice. The serum concentration of GDF15 in mice on HFD was around 120 pg/ml, higher than that in mice on NCD (65 pg/ml) (see Fig. 1 below). According to a published study (PMID: 28475119) that used the same ELISA kit, mice that underwent weight loss had a GDF15 concentration in the range 300–600 pg/ml. Plasma concentrations of GDF15 associated with cachexia in this model were at least twice as high as those in mice on HFD. Although GDF15 decreases food intake, the plasma concentration–effect relationship remains to be elucidated. We expect that the pharmacological effects of GDF15 on body weight and food intake may be caused by very high concentrations of the protein. We suggest that the physiological concentration of GDF15 (increase by HFD or the difference in the concentration of GDF15 between WT and GDF15KO) may not be high enough to reduce food intake and body weight. Unfortunately, recent publications on GDF15-GFRAL did not determine the plasma concentrations that effectively suppress food intake in rodents and primates. We have added a description of this discrepancy in the Discussion section of the revised manuscript (pages 20-21, lines 444-452).

Fig. 1. Serum GDF15 levels of MacWT and MacHO mice after normal chow diet for 14 weeks or high-fat diet for 8 weeks. Data are expressed as means \pm SEM and are representative of three independent experiments (n=4 mice per group). *p<0.05 (two-tailed Student's t-test), # non-significant.

Q3) As the authors already discussed, recent studies suggest that GDF15-GFRAL axis is a novel regulator of metabolic homeostasis, primarily affecting food intake and resulting in body weight changes. Although these effects are mainly mediated by receptors expressed in the brain, I agree with the authors that different mechanisms might be responsible for its action in the periphery. Because the authors used global GDF15 KO mice rather than using macrophage-specific KO mice, it is possible there are additional defects in these animals. For instance, did the authors check food intake in their experiments?

A3) Several studies have shown that food intake does not differ between GDF15-deficient and control wild-type mice. For example, Mazagova M. et al (PMID: 23986522) showed that body weights and food intake were similar between GDF15 KO and WT mice. Tsai VW et al. (PMID: 23468844) showed that male $MIC1^{-/-}$ mice have metabolic activity and food intake similar to those of syngeneic control mice at the age of 13–16 weeks. Consistent with the results of previous studies, we observed that there was no difference in food intake between WT and GDF15 KO mice prior to the age of 16 weeks.

Q4) In some experiments, HOMA-IR was presented as an index of insulin resistance (e.g. Fig. 8). Because HOMA-IR is a less informative index for measuring the degree of insulin resistance, I wonder if the authors tested ITT in these models.

A4) We thank the reviewer for this constructive comment and suggestion. We examined and placed the GTT and ITT results in supplemental Figure 6C and D. We have revised the supplemental figure 6 legend in the “Supplemental Fig” part accordingly. These data show an increase in insulin sensitivity in *ob/ob* mice injected with GDF15. And we have included these findings in the revised manuscript (Results section of the revised manuscript, page 16, lines 364-366).

Q5) Line 113: “reduction in reduced basal” -> “reduced basal”

A5) We changed “reduction in reduced basal” to “reduced basal”, as the reviewer suggested (Results section of the revised manuscript, page 6, line 116).

Q6) Figure 2(a) legend: It is written that HFD was administered for 8 weeks, whereas the figure shows the data for more than 10 weeks after starting HFD. Please make clear for the duration of HFD.

A6) We corrected “8 weeks” to “10 weeks” in the figure legend (Figure legends section of the revised manuscript, page 39, line 862).

Q7) Figure 3(a, e-h) legend: Please indicate the duration of IL-4/RSG treatment.

A7) We indicated the treatment duration in the figure legend (Figure legends section of the revised manuscript, page 40, line 878, line 883).

Reviewer #2 (Remarks to the Author):

Review for the manuscript entitled 'Reduced oxidative capacity in macrophages results in systemic insulin resistance and its reversal with GDF15 treatment by Jung et al.

In this study, Jung and colleagues generate mice with a myeloid cell-specific deletion of CR6-interacting factor 1 (Crif1) and demonstrate that these mice exhibit reduced mitochondrial oxidative phosphorylation, in addition to a shift towards a M1-like polarization of macrophages, whole-body insulin resistance and local adipose tissue inflammation i.e. overall an unhealthy metabolic phenotype due to the deletion of Crif1 from macrophages. The authors further report that GDF15 expression is reduced in macrophages that harbor this Crif1 deletion, and that this is concomitant with impaired mitochondrial oxidative capacity. Conversely, treatment with the TZD rosiglitazone, and interleukin-4 (IL-4), appeared to reverse this phenotype and enhance macrophage GDF15 expression. Finally, the authors show that the macrophage GDF15 up-regulation (as stimulated by rosiglitazone and IL-4), enhances the oxidative capacity of the cell, causing it to switch from an M1-like state, to an M2-like state of polarization; this then reverses insulin resistance in HFD-fed mice and *ob/ob* mice that are null for Crif1 in macrophages. The authors conclude that reduced oxidative capacity specifically in macrophages is the key determinant of adipose tissue inflammation and whole-body insulin resistance, and that this can essentially be reversed by means of enhanced GDF15 levels in these macrophages.

Overall, this is a good study that is very technically sound and contributes to the current literature on GDF15. However, there are major concerns regarding the novelty of the study and the fact that an almost identical mechanism and study has previously been published by the authors in 2016, albeit in a different cell type, skeletal-muscle cells.

Q1) The authors show that macrophage GDF15 levels are reduced with impaired mitochondrial oxidative function. However, in the literature, GDF15 has been shown to function as a 'mitochondrial stress marker', and levels are in fact increased under conditions of mitochondrial dysfunction and mitochondrial disease. For instance, this has been shown by numerous studies: (1) Fujita Y et al. (Secreted growth differentiation factor 15 as a potential biomarker for mitochondrial dysfunctions in aging and age-related disorder. *Geriatr Gerontol Int* 2016), (2) Montero R et al. (GDF-15 is elevated in children with mitochondrial disease and is induced by mitochondrial dysfunction. *PLoS One* 2016), and (3) Koene S et al. (Serum GDF15 levels correlate to mitochondrial disease severity and myocardial strain, but not disease progression in adult *m.3242A>G* carriers. *JIMD Rep* 2015). As such, how do the authors explain this discrepancy with the literature? High GDF15 levels have been related to mitochondrial dysfunction in several publications. Here the authors state that high GDF15 levels improve mitochondrial function, and that GDF15 is required for oxidative metabolism of the M2-macrophage. The authors also state (in the Introduction), that GDF15 is a macrophage-regulating autocrine and paracrine signaling factor that promotes anti-inflammatory responses in white adipose tissue. This discrepancy with the literature is concerning and should be addressed.

A1) As such, how do the authors explain this discrepancy with the literature?

As the reviewer indicated, several groups have shown that plasma GDF15 levels are elevated in mice and humans with mitochondrial OxPhos dysfunction. The elevated plasma GDF15 levels in mice and humans with mitochondrial OxPhos dysfunction may result from higher secretion of GDF15 in affected tissues, including skeletal muscle⁵⁰. However, induction and secretion of GDF15 in cells with compromised OxPhos function is largely dependent on cell type. **[Redacted]**. Based on these observations, we speculate that GDF15 secretion in Crif1-deficient macrophages is a cell type-specific response to mitochondrial stress. We have added a description on this discrepancy to the Discussion section of the revised manuscript (page 20, lines 425-434).

[Redacted]

[Redacted]

Q2) The authors have already published what appears to be a parallel study with an identical mechanism of GDF15 already in 2016 in JCB (Chung HK: Growth differentiation factor 15 is a myomitokine governing systemic energy homeostasis). Here, the authors deleted Crif1 from the muscle and showed an identical mechanism, whereby elevated GDF15 from muscle improved insulin sensitivity and oxidative metabolism in ob/ob mice. The novelty of this current study is therefore greatly reduced, as the authors have already published this mechanism and phenotype, albeit simply performed the study in a different cell type.

A2) The major observation in the JCB paper was that GDF15 is a relevant mitokine derived from skeletal muscle. In this study, we showed that GDF15 is induced by stimulation with a PPAR γ agonist and Th2 cytokine; this observation is distinct and we do not imply that it acts as a mitokine in this condition. We believe that the major novelty of this manuscript is 2-fold: 1) We provide experimental confirmation of the hypothesis that macrophages with compromised OxPhos promote insulin resistance with adipose inflammation. 2) We demonstrate that GDF15 links macrophage polarization phenotypes under impaired OxPhos conditions. Although we showed previously that GDF15 has therapeutic potential to improve insulin sensitivity, this does not significantly decrease the novelty of the current study because here we focused on the action of GDF15 in macrophages.

Q3) In Figure 4f, the OCR values should be plotted as absolute values, not baselined to palmitate, as this can exaggerate the differences observed. The authors should use the absolute OCR values (pmol/min), as they have done for their other oxidative respiration graphs throughout the manuscript.

A3) We changed the OCR values (pmol/min) in Fig. 4f of the "Main Fig" part, as the reviewer suggested.

REVIEWERS' COMMENTS:

Reviewer #1 (Remarks to the Author):

The authors have adequately revised the manuscript to address my concerns. It is acceptable for publication in Nature Communications.

Reviewer #2 (Remarks to the Author):

The authors have addressed some of the major points raised in a satisfactory manner. The key issue remains the relative novelty of the findings compared to their previously published data on the topic, which make the present contributions in this paper rather incremental.